# Identification of a diarylpentanoid-producing polyketide synthase revealing an unusual biosynthetic pathway of 2-(2-phenylethyl) chromones in agarwood

Xiao-Hui Wang[1,7], Bo-Wen Gao[1,5,7], Yu Nakashima[2,7], Takahiro Mori [3], Zhong-Xiu Zhang[1,6], Takeshi Kodama[2], Yuan-E Lee[2], Ze-Kun Zhang[1], Chin-Piow Wong[2], Qian-Qian Liu[2], Bo-Wen Qi[1], Juan Wang[1], Jun Li [1], Xiao Liu [1], Ikuro Abe [3], Hiroyuki Morita [2✉], Peng-Fei Tu [1,4✉] & She-Po Shi [1✉]

2-(2-Phenylethyl)chromones (PECs) are the principal constituents contributing to the distinctive fragrance of agarwood. How PECs are biosynthesized is currently unknown. In this work, we describe a diarylpentanoid-producing polyketide synthase (PECPS) identified from *Aquilaria sinensis*. Through biotransformation experiments using fluorine-labeled substrate, transient expression of PECPS in *Nicotiana benthamiana*, and knockdown of *PECPS* expression in *A. sinensis* calli, we demonstrate that the $C_6$–$C_5$–$C_6$ scaffold of diarylpentanoid is the common precursor of PECs, and PECPS plays a crucial role in PECs biosynthesis. Crystal structure (1.98 Å) analyses and site-directed mutagenesis reveal that, due to its small active site cavity (247 Å$^3$), PECPS employs a one-pot formation mechanism including a "diketide-CoA intermediate-released" step for the formation of the $C_6$–$C_5$–$C_6$ scaffold. The identification of PECPS, the pivotal enzyme of PECs biosynthesis, provides insight into not only the feasibility of overproduction of pharmaceutically important PECs using metabolic engineering approaches, but also further exploration of how agarwood is formed.

[1] Modern Research Center for Traditional Chinese Medicine, Beijing University of Chinese Medicine, Beijing 100029, People's Republic of China. [2] Institute of Natural Medicine, University of Toyama, Sugitani-2630, Toyama 930-0194, Japan. [3] Graduate School of Pharmaceutical Sciences, The University of Tokyo, 7-3-1 Hongo, Bunkyo-ku Tokyo 113-0033, Japan. [4] State Key Laboratory of Natural and Biomimetic Drugs, School of Pharmaceutical Sciences, Peking University, Beijing 100191, People's Republic of China. [5] Present address: Baotou Medical College, Baotou 014060, People's Republic of China. [6] Present address: Institute of Chinese Materia Medica, China Academy of Chinese Medical Sciences, Beijing 100700, People's Republic of China. [7] These authors contributed equally: Xiao-Hui Wang, Bo-Wen Gao, Yu Nakashima. ✉email: hmorita@inm.u-toyama.ac.jp; pengfeitu@163.com; shishepo@163.com

Agarwood, also known as aloeswood, gaharu, eaglewood, jinkoh and Chenxiang, is a quite expensive resinous wood gradually formed by Thymelaeaceae plants in response to abiotic and/or biotic stresses such as fungal infection, hurricane wounds, and insect bites[1,2]. Agarwood is widely used in Oriental medicine as digestive, sedative, and antiemetic agents for the treatment of stomachache, emesis, and insomnia[3]. In addition, agarwood is an important aromatic ingredient in perfumes, incense ceremonies, and craft production because of its unique and pleasant fragrance[2–5]. However, as a pathological product of Thymelaeaceae plants, high-quality agarwood is extremely rare due to its infrequent occurrence in natural environments and long forming period. With the ever-increasing market demand for agarwood, the ecology of wild agarwood-producing plants has been destroyed due to overharvesting and excessive logging[6–8]. Accordingly, wild agarwood-producing plants, such as *Aquilaria* spp. and *Gyrnops* spp., were included in Appendix II of the Convention of International Trade in Endangered Species of Wild Fauna and Flora (CITES). Consequently, artificial cultivation of agarwood-producing plants has recently caused widespread concern in China, Indonesia, Cambodia, Thailand, Vietnam, and some other countries, and a variety of artificial agarwood induction techniques including fungal inoculation, physical damage such as trunk pruning and burning-chisel drilling, and chemical elicitation have been developed[9–13]. However, the quality and yield of agarwood produced using these artificial approaches are less than satisfactory. Hence, it is of great interest to unveil the molecular mechanism to thereby artificially manipulate the process of agarwood formation. It has been demonstrated that the formation of agarwood is characteristically accompanied by the synthesis and accumulation of sesquiterpenoids and 2-(2-phenylethyl)chromones (PECs), which are the principal constituents contributing to the important biological activities and the pleasant fragrance of agarwood[14–17]. To decipher the molecular mechanism of agarwood formation, great efforts have been made in the past decade to explore the biosynthesis of these two classes of chemicals in agarwood. Although the enzymes involved in the biosynthetic pathway of agarwood sesquiterpenoids have been reported[18–24], the biosynthesis of PECs currently remains completely unknown.

PECs are a subgroup of chromones that characteristically bear a phenylethyl group at the C-2 position. Since the first natural PEC, flindersiachromone, was elucidated in 1976[25], nearly 250 PECs have been isolated exclusively from agarwood[14–17,26–29]. In the Chinese pharmacopeia, PECs were legally assigned as diagnostic constituents for evaluating the quality of agarwood. Based on the core structures, PECs could be classified into three groups: flindersiachromones (FDC-type), oxidoagarochromones (OAC-type), and agarotetrolchromones (ATC-type) (Fig. 1a). The flexible variations of the substituents on the core structures and the complex stereochemical configurations occurring in OAC- and ATC-type PECs have greatly expanded the structural diversity and chemical complexity of PECs. Recently, PEC dimers[30–34], trimers[34–36], and PEC-sesquiterpene hybrids[37–39] have also been reported. Remarkably, recent studies have revealed that PECs possess a broad spectrum of bioactivities, such as neuroprotective[40], acetylcholine esterase inhibitory[41], anti-inflammatory[42,43], and antibacterial[44]. In particular, PECs GYF-17 and GYF-21 were demonstrated to be promising inhibitors of the STAT1/3 signaling pathway[45,46], and 5-hydroxy-2-(2-phenylethyl)chromone (5-HPEC) was reported to be a selective antagonist of the 5-HT$_{2B}$ receptor[47,48]. Thus, elucidation of the biosynthetic pathway of PECs would not only inspire the biomimetic or enzymatic synthesis of structurally diverse and chemically complex PECs but also benefit the industries that depend on agarwood.

In this work, we demonstrated that PECs in agarwood are biosynthesized from a common precursor featuring a $C_6–C_5–C_6$ skeleton produced by a diarylpentanoid-producing polyketide synthase, hereafter named 2-(2-phenylethyl)chromone precursor synthase (PECPS). Crystal structure analyses suggested that PECPS utilizes unique catalytic machinery with the formation and release of a diketide-CoA intermediate to form the $C_6–C_5–C_6$ scaffold, which has not been previously reported in other known type III polyketide synthases (PKSs).

## Results and discussion

**Cloning PECPS from *Aquilaria sinensis*.** We have previously reported structurally diverse PECs produced by *A. sinensis* calli in 150 mM NaCl treated conditions (Supplementary Fig. 1), as well as RNA sequencing of these PEC-producing calli treated with NaCl and PEC-nonproducing healthy calli[49]. Considering the high similarity of the backbone structures of PECs to those of flavonoids biosynthesized by type III PKSs, this type of enzyme might also play an important role in the biosynthesis of PECs. Accordingly, comprehensively screening the transcriptomic dataset led to the identification of five *PKS* genes with upregulated expression. Initially, we focused on the cloning and functional analysis of *AsCHS1*, which showed the most significantly upregulated expression among the five candidates. However, *AsCHS1* cloned from *A. sinensis* always contains an intron and cannot be heterologously expressed in *Escherichia coli* even after removal of the intron. Therefore, we switched to identify the catalytic functions of the other four candidate genes, which resulted in the identification of one chalcone-producing PKS (AsCHS) and two pyrone-producing PKSs (AsPKS1 and AsPKS2)[50]. The last remaining candidate gene encodes a PKS (PECPS) sharing high amino acid sequence identity with the other four PKSs, as well as the highest 93% similarity with AsCHS1, and this gene maintains the Cys-His-Asn catalytic triad conserved in all known type III PKSs (Supplementary Fig. 2). Phylogenetic analysis revealed that PECPS was grouped into the class of plant non-chalcone synthases (Supplementary Fig. 3). Fortunately, the full-length PECPS protein could be easily expressed in *E. coli* as a fusion protein with a hexahistidine tag at the N-terminus. The purified recombinant PECPS migrated as a single band with a molecular weight of approximately 40 kDa on SDS–PAGE, which is consistent with the calculated molecular weight of 43 kDa (Supplementary Fig. 4a). In contrast, size-exclusion chromatography indicated a homodimeric protein with a molecular weight of approximately 86 kDa (Supplementary Fig. 4b).

**Determination of the enzymatic activity of PECPS.** Considering that PECs contain a phenylethyl moiety, 4-hydroxyphenylpropionyl-CoA, rather than *p*-coumaroyl-CoA, was deliberately selected as the starter substrate for the reaction with malonyl-CoA, a canonical chain extender for PKSs, and the recombinant PECPS. The reaction specifically generated a diarylheptanoid $C_6–C_7–C_6$ scaffold assigned as tetrahydrobisdemethoxycurcumin (**1**) (Fig. 1a, b, Supplementary Fig. 5, and Supplementary Table 1), since diarylheptanoid derivatives have never been isolated from agarwood. The construction of a $C_6–C_5–C_6$ scaffold by coincubation of benzoyl-CoA, malonyl-CoA, and 4-hydroxyphenylpropionyl-CoA was attempted. As expected, in addition to the production of **1**, an unknown product (**2**) with a deduced molecular formula of $C_{17}H_{16}O_3$ (*m/z* 267.1027 [M - H]$^-$, calc. for $C_{17}H_{15}O_3$ 267.1027) was generated (Fig. 1b). Through NMR investigation, product **2** was elucidated as 5-(4-hydroxyphenyl)-1-phenylpentane-1,3-dione (Supplementary Fig. 6 and Supplementary Table 1), a diarylpentanoid with a $C_6–C_5–C_6$ backbone. Interestingly, diarylpentanoids have been isolated from agarwood[51]. Further qRT-PCR (Supplementary Fig. 7) and western blot analysis using a specific

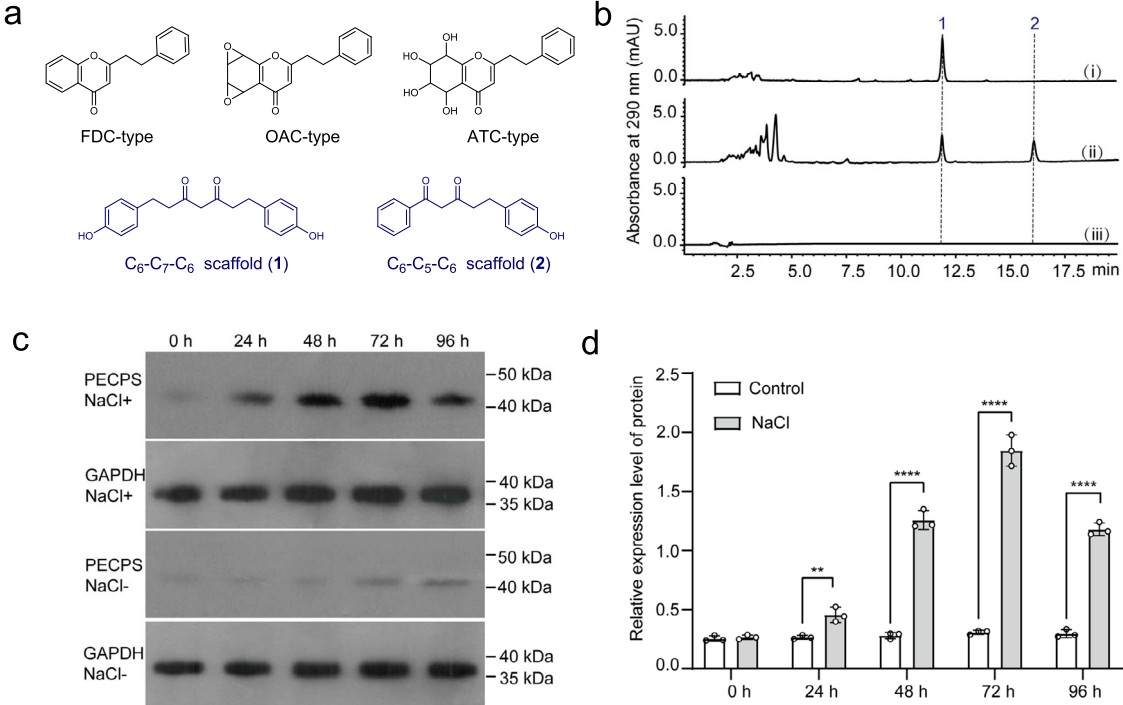

**Fig. 1 Representative 2-(2-phenylethyl)chromone skeletons and the identification of PECPS. a** The core structures of three groups of 2-(2-phenylethyl) chromones and enzymatic products **1** and **2**. **b** HPLC chromatograms for the formation of the $C_6$–$C_7$–$C_6$ scaffold of tetrahydrobisdemethoxycurcumin (**1**) from 4-hydroxyphenylpropionyl-CoA and malonyl-CoA by PECPS (i), The formation of the $C_6$–$C_5$–$C_6$ scaffold of 5-(4-hydroxyphenyl)-1-phenylpentane-1,3-dione (**2**) from benzoyl-CoA, malonyl-CoA, and 4-hydroxyphenylpropionyl-CoA by PECPS (ii) and boiled PECPS (iii). **c** Western blot analysis of the PECPS protein expression levels in 150 mM NaCl-treated *A. sinensis* calli (NaCl+) and healthy *A. sinensis* calli (NaCl-) at different time points. GAPDH was used as internal reference. **d** Western blot quantification of the relative expression levels of PECPS by ImageJ and normalization to the protein levels of GAPDH. Data represent the mean ± SD ($n = 3$). Statistical analysis was performed with unpaired two-tailed Student's t-tests. Significance is shown with asterisks: ****$p < 0.0001$, ***$p < 0.001$, **$p < 0.01$. Exact $p$-values and source data are provided in the source data file. Total protein was extracted from 100 mg of calli using Plant Total Protein Extraction Reagent (Invent, USA) according to the manufacturer's instructions. The protein expression levels of PECPS were analyzed by Western blotting with an anti-PECPS antibody prepared in rabbits.

antibody against PECPS prepared in rabbits (Fig. 1c, d) revealed that the expression level of PECPS in NaCl-stressed cells notably increased, suggesting the possible involvement of PECPS in the biosynthesis of PECs. Accordingly, we speculated that PECs are biosynthesized from a common precursor featuring a $C_6$–$C_5$–$C_6$ skeleton produced by PECPS, differing from the biosynthesis of flavonoids derived from a common precursor of naringenin chalcone which is generated via Claisen-type cyclization of a linear tetraketide intermediate produced by chalcone synthase (CHS) from *p*-coumaroyl-CoA and three malonyl-CoAs[52,53].

**Biotransformation of diarylpentanoid-precursor into PECs.** To verify the aforementioned hypothesis that diarylpentanoid is the common precursor of PECs, substrate-feeding experiments were performed. First, a putative innate-substrate, 5-(4-hydroxyphenyl)-1-phenylpentane-1,3-dione (**2**), was fed to NaCl-stressed *A. sinensis* cells, and its metabolites were carefully analyzed by LC–MS. In the presence of substrate **2**, the accumulation of PECs (**P1**–**P5**) in substrate-fed cells visibly increased (Fig. 2a), suggesting that **2** might be converted to PECs. Next, the fluorine-containing nonphysiological substrate 5-(4-F-phenyl)-1-phenylpentane-1,3-dione (**F-2**) was fed to NaCl-stressed *A. sinensis* cells. As a result, 10 fluorinated PECs (**C01**–**C10**) were successfully obtained by LC–MS guided separation and purification (Fig. 2b and Supplementary Fig. 8), and the structures of **C01**–**C06** were unequivocally determined by extensive spectroscopic techniques, including ¹H-NMR, ¹³C-NMR and HRESI-MS (Supplementary Figs. 9–14 and Supplementary Tables 2 and 3).

Unfortunately, attempts to elucidate the structures of compounds **C07**–**C10** by NMR techniques were complicated by their small yields. However, the HRESI-MS spectra of **C07**–**C10** indicated the characteristic cleavage of PECs, which tends to yield a fluorine-containing benzyl fragment, allowing us to tentatively determine the structures of **C07**–**C10** (Supplementary Figs. 15–18). Interestingly, the fluorinated PECs (**C01**–**C10**) are the analogs in which the F atoms replaced the OH groups of the natural PECs. Honda and coworkers proposed that OAC-type PECs are the key precursors of FDC- and ATC-type PECs[54]. However, feeding experiments demonstrated that the $C_6$–$C_5$–$C_6$ scaffold is a key precursor to form flindersiachromone (**C01**), the simplest FDC-type PEC, and **C01** then serves as a key intermediate to form structurally diverse FDC-type PECs (**C02**–**C09**) via hydroxylation and subsequent *O*-methyltransfer reactions (Fig. 2b). Furthermore, production of the highly oxygenated tetrahydrochromone **C10** suggested that **C01** (FDC-type) might be the precursor of OAC-type PECs, and then the OAC-type PECs are converted to ATC-type PECs by the opening of the epoxy rings by nonenzymatic reactions and/or epoxide hydrolases (Fig. 2c). In fact, the products produced from the opening of the epoxy ring could be readily detected from a methanol aqueous solution of oxidoagarochromone C (Supplementary Fig. 19), supporting the possibility of the nonenzymatic conversion of OAC-type PECs into ATC-type PECs. However, a portion of natural ATC-type PECs[26,31,32] contain *cis*-6,7-diol fragments, which are theoretically difficult to be generated from the spontaneous opening of the epoxy ring, suggesting the possibility that

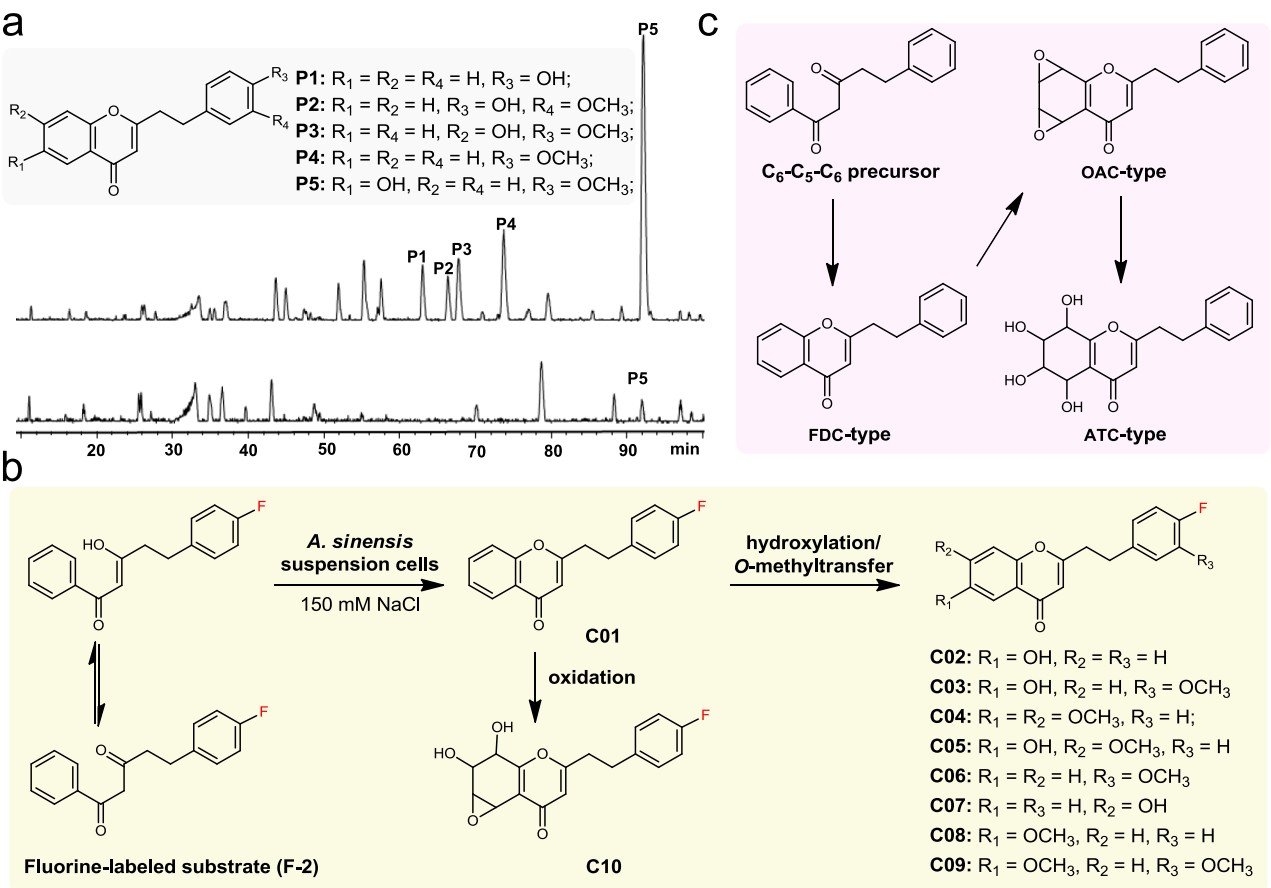

**Fig. 2 Biotransformation of the diarylpentanoid-precursor to structurally varied PECs. a** HPLC chromatogram of PECs produced by suspension cells fed with 5-(4-hydroxyphenyl)-1-phenylpentane-1,3-dione (**2**). **b** Conversion of fluorine-labeled substrate (**F-2**) to structurally varied fluorinated PECs by suspension cells. **c** Proposed biosynthetic pathway of the three types of PEC skeletons.

an enzyme, such as an epoxide hydrolase, catalyzes the stereo-selective ring-opening reaction.

**The in vivo biochemical function of PECPS**. PECPS was expressed in *N. benthamiana* leaves via *Agrobacterium*-mediated transient expression (Supplementary Fig. 20a), and the metabolites in the *N. benthamiana* leaves were carefully analyzed by LC–MS after the plants were growing in a chamber for another 5 days. As a result, the *N. benthamiana* leaves expressing PECPS accumulated a unique compound identified as 5-hydroxy-1,7-bis(4-hydroxyphenyl)heptan-3-one (**1R**), corresponding to the reduced product of **1**, by LC–MS/MS analysis. In contrast, accumulation of this compound was not observed in the wild type *N. benthamiana* leaves (Fig. 3a, b). Furthermore, benzoyl-CoA feeding experiment on *N. benthamiana* leaves expressing PECPS led to the accumulation of 2-(4-hydroxyphenylethyl)-4*H*-chromen-4-one (**P1**), corresponding to the cyclized product of 5-(4-hydroxyphenyl)-1-phenylpentane-1,3-dione (**2**), which was unambiguously identified by comparisons of its retention time on the column and MS data with those of the standard compound (Fig. 3a, b). Further infiltration of 5-(4-hydroxyphenyl)-1-phenylpentane-1,3-dione (**2**) and 1,5-diphenylpentane-1,3-dione with the C6-C5-C6 scaffold into the wild type *N. benthamiana* leaves revealed that these compounds were converted to PECs, 2-(4-hydroxyphenylethyl)-4*H*-chromen-4-one (**P1**) and 2-phenylethyl-4*H*-chromen-4-one, respectively (Supplementary Fig. 21), suggesting that PECPS could generate **2** in wild type *N. benthamiana* leaves, and then, **2** was further cyclized by oxidase(s) indigenous to *N. benthamiana* to form the PEC scaffold (**P1**). In addition, the expression level of *PECPS* in *A. sinenesis* calli was knocked down using the

RNAi method (Supplementary Fig. 20b), and PECs in *PECPS* knockdown calli were carefully analyzed and quantified by LC–MS. Compared with those of the calli treated with 150 mM NaCl alone, the contents of PECs in the *PECPS* knockdown calli (also treated with 150 mM NaCl) were dramatically decreased (Fig. 3c, d and Supplementary Fig. 22), confirming that PECPS is a pivotal enzyme involved in the biosynthesis of PECs in agarwood.

**Structural basis for the PECPS catalytic mechanism**. To explore the underlying mechanism for the formation of the C6−C5−C6 scaffold by PECPS, we solved the crystal structure of PECPS at 1.98 Å resolution. The crystal structure of PECPS showed typical homodimeric type III PKS folding (Supplementary Fig. 23). The Cys-His-Asn catalytic triad was also sterically well conserved in PECPS, in a location and orientation very similar to those of the other type III PKSs (Supplementary Fig. 24). However, a comparison of the structures of PECPS and naringenin chalcone-producing *Medicago sativa* CHS (MsCHS) suggested that PECPES may differ from MsCHS by the so-called 'coumaroyl-binding pocket' (pocket A), which is thought to lock the aromatic moiety of the intermediates[55]. In these comparisons, the equivalent pocket A in PECPS could be tapered off the "main" catalytic pocket via a large steric shift from MsCHS's Ser338 to PECPS's Phe340 (Fig. 4a, b), leading to a smaller cavity (247 Å³) compared to those of the apo- and naringenin-complexed structures of MsCHS (PDB ID: 1BI5; 742 Å³ and PDB ID: 1CGK; 754 Å³, respectively). Despite this, docking simulation predicted that the volume of the smaller PECPS cavity is sufficient to accommodate the 4-hydroxyphenylpropionyl-β-diketide unit.

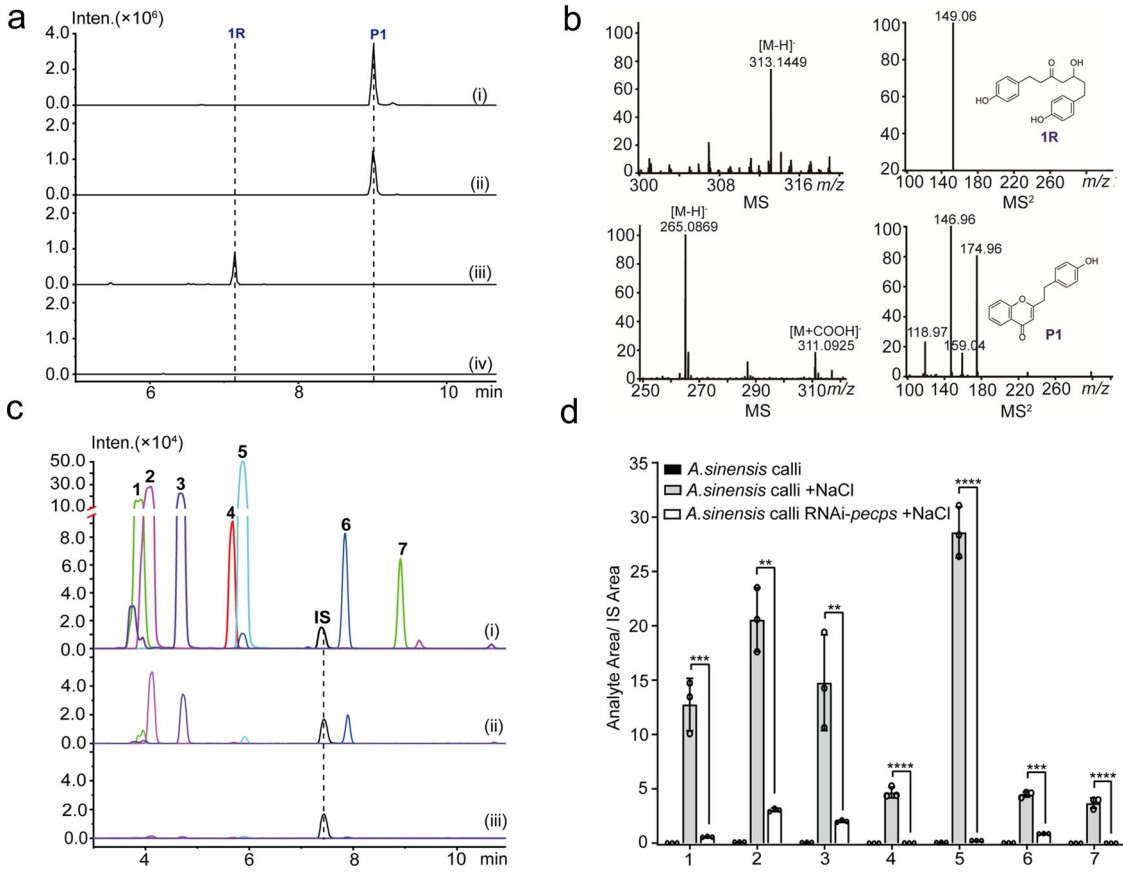

**Fig. 3 Determination of the in vivo biochemical function of PECPS by transient expression of PECPS in *N. benthamiana* and knockdown expression of PECPS in *A. sinensis calli*. a** MS chromatograms (EIC) of 5-hydroxy-1,7-bis(4-hydroxyphenyl)heptan-3-one (**1R**) and 2-(4-hydroxyphenylethyl)-4*H*-chromen-4-one (**P1**) accumulated in *N. benthamiana* leaves: (i) standard of **P1**; (ii) leaves infiltrated with benzoyl-CoA and *Agrobacterium* harboring pCAMBIA1300-35S-*pecps*; (iii) Leaves infiltrated with *Agrobacterium* harboring pCAMBIA1300-35S-*pecps*; (iv) leaves infiltrated with *Agrobacterium* harboring pCAMBIA1300-35S vector. **b** MS and MS² spectra of **1R** and **P1**. **c** MS chromatograms (EIC) of the major PECs **1–7** (structures could be found in Supplementary Fig. 22) accumulated in *A. sinensis* calli treated by different methods: (i) treated with 150 mM NaCl; (ii) knockdown PECPS expression and treated with 150 mM NaCl; (iii) healthy calli. **d** The relative contents (1,5-diphenylpentane-1,3-dione was implemented as internal standard, **IS**) of PECs **1–7** accumulated in different *A. sinensis* calli. Data represent the mean ± SD (*n* = 3). Statistical analysis was performed with unpaired two-tailed Student's t-tests. Significance is shown with asterisks: ****p < 0.0001, ***p < 0.001, **p < 0.01. Exact *p*-values and source data are provided in the source data file.

To investigate whether the small pocket A is required for enzyme activity, we mutated Phe340 to the bulkier residue Trp and found that the mutant retained its **1**-producing activity and binding ability with 4-hydroxyphenylpropionyl-CoA at levels comparable to those of the wild type in HPLC and isothermal titration calorimetry (ITC) experiments (Fig. 4h, i and Supplementary Table 4). Furthermore, the crystal structure of the F340W mutant indicated a similar size and shape of the main catalytic cavity to that of the wild type, suggesting that pocket A is not required for substrate and intermediate binding during the enzyme reaction (Fig. 4d).

Interestingly, the active site architecture of PECPS is apparently different from that of curcuminoid synthase (CUS), a plant-specific type III PKS that catalyzes the one-pot formation of bisdemethoxycurcumin from the condensation of two *p*-coumaroyl-CoAs and one malonyl-CoA[56]. In PECPS, pocket B corresponding to pocket C in the apo-structure of CUS (PDB ID: 3ALE), consisting of residues Tyr207, Thr209, Arg217, Thr218, Met265, Arg271, Leu272, Asp273, and Gly274, is absent (Fig. 4a, c), resulting in a cavity volume (247 Å³) that is 2.5 times smaller than that of CUS (642 Å³). The significant difference between the catalytic cavities of PECPS and CUS is principally attributed to the steric volumes and placements of Asn199, Thr213, and Pro267 in PECPS, which form a constricting loop to

cut off auxiliary pocket B from the overhead main catalytic pocket (Supplementary Fig. 25). However, in light of the reported CUS catalytic mechanism, where an aromatic portion of the *p*-coumaroyldiketide acid intermediate is locked into pocket C to accept the second extender substrate, *p*-coumaroyl-CoA[57], the side chain of Asn199, located as the gatekeeper of pocket B, in PECPS may adopt other rotameric configurations during the catalytic reaction to form a larger catalytic cavity consisting of pocket B and the overhead main catalytic pocket similar to that in CUS. To investigate the role of pocket B in the catalytic reaction of PECPS, Ala210, located in the vicinity of pocket B, was substituted with a bulkier residue (Glu210), which possesses a protruding side chain that may extend into pocket B. However, the A210E mutant did not lose its catalytic activity and exhibited mostly the same **1**-producing activity as that of the wild type in the HPLC experiment (Fig. 4h, i). The binding affinity ($K_D$) of the A210 mutant was estimated to be 40.12 ± 4.52 μM with respect to 4-hydroxyphenylpropionyl-CoA, which was slight weaker than that of wild type PECPS ($K_D$ = 18.57 ± 1.37 μM) (Supplementary Table 4). The crystal structure of the A210E mutant enzyme revealed that the A210E substitution eliminated pocket B observed in the wild type PECPS structure by occupying a space corresponding to pocket B with the Glu210 side chain. Other significant conformational changes between the wild type and

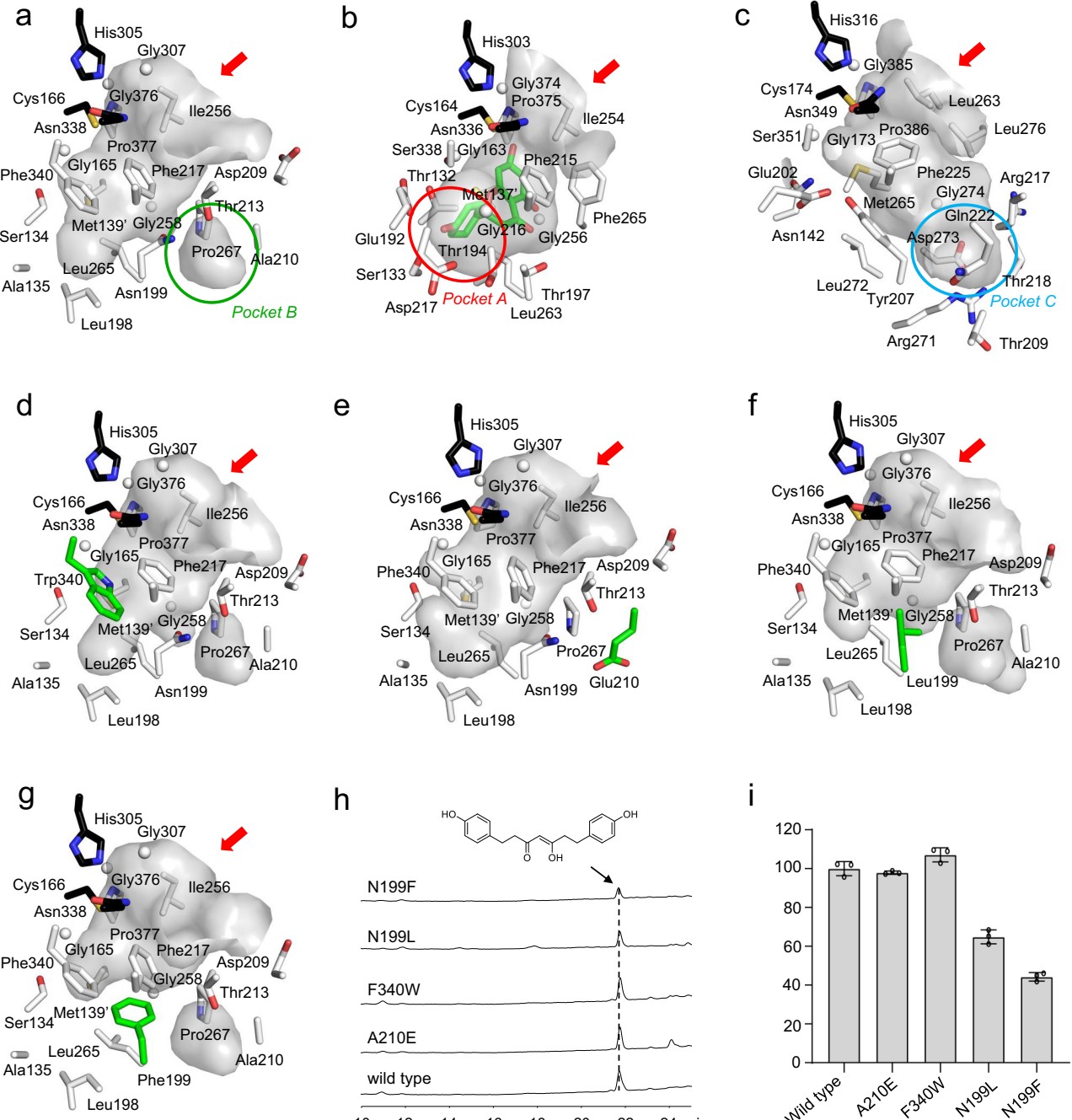

**Fig. 4 Comparison of the active site topologies in the crystal structures and the activities of various PECPS mutants. a** Wild type PECPS. **b** MsCHS. **c** CUS. **d** PECPS F340W. **e** PECPS A210E. **f** PECPS N199L. **g** PECPS N199F. **h** HPLC chromatograms of the products obtained from the enzymatic reactions of wild type PECPS and its mutants using 4-hydroxyphenylpropionyl-CoA and malonyl-CoA as substrates. Chromatograms were recorded at 295 nm. **i** Relative activities for the formation of tetrahydrobisdemethoxycurcumin (**1**) between wild type PECPS and its mutants. Activities are percentages of tetrahydrobisdemethoxycurcumin (**1**) production relative to that of wild type PECPS. Data represent the mean ± SD (n = 3). PDB ID: **a** 7FFA. **b** 3ALE. **c** 1CGK. **d** 7FFI. **e** 7FFC. **f** 7FFH. **g** 7FFG.

A210E mutant were not observed in their crystal structures (Fig. 4e). The docking simulation also showed no critical movement of the side chains, including the apparent rotation of Asn199, even though the substrate and intermediate were located in the active site cavity (Fig. 5). These results suggested that the PECPS catalytic reaction is independent of pocket B.

We also constructed Asn199 mutants in which Asn was substituted with Leu199 and Phe199 to investigate the role of Asn199 in enzyme activity. Interestingly, the activities of N199L and

N199F decreased by 35% and 56%, respectively, compared with that of wild type PECPS (Fig. 4h, i). ITC analyses indicated that the N199L mutant possessed a binding affinity of a $K_D$ value of 120.40 ± 10.78 μM with respect to 4-hydroxyphenylpropionyl-CoA, which was 6.5 times weaker than that of wild type PECPS (18.57 ± 1.37 μM). Furthermore, the N199F mutant possessed significantly lower binding affinity as compared with that of the N199L mutant ($K_D$ value is close to 1000 μM) (Supplementary Table 4). Remarkably, the crystal structures of N199L (Fig. 4f) and

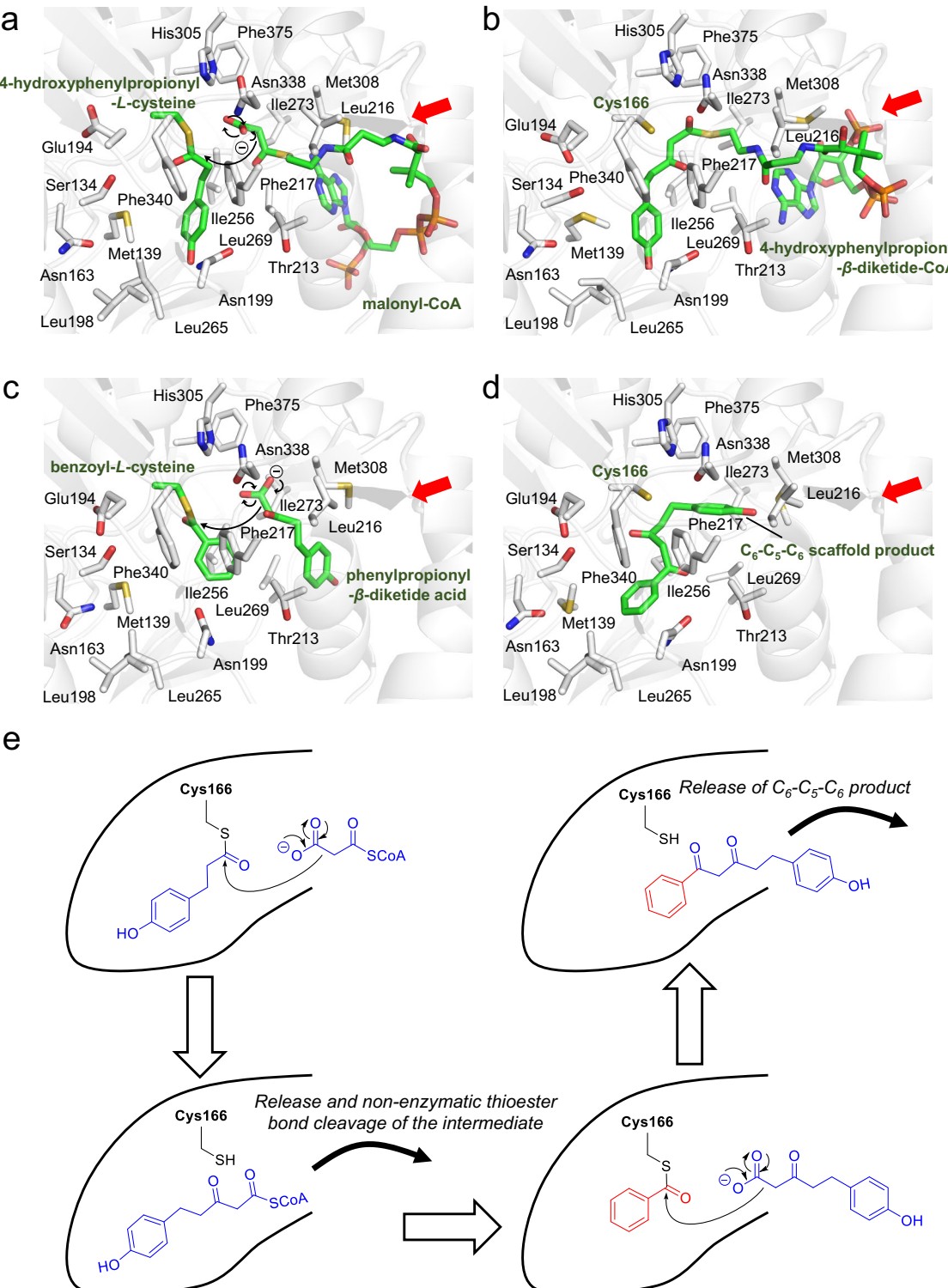

**Fig. 5 Proposed mechanism of the formation of the C₆–C₅–C₆ scaffold by PECPS. a** Decarboxylative condensation of malonyl-CoA with the enzyme-bound 4-hydroxyphenylpropionyl unit. **b** Formation of the 4-hydroxyphenylpropionyl-β-diketide-CoA intermediate. **c** Decarboxylative condensation of phenylpropionyl-β-diketide acid nonenzymatically derived from phenylpropionyl-β-diketide-CoA with the enzyme-bound benzoyl unit. **d** Formation of the C₆–C₅–C₆ scaffold product. **e** Schematic representation of the proposed mechanism of the formation of the C₆–C₅–C₆ scaffold by PECPS. The entrances to the catalytic cavities are indicated with red arrows.

N199F (Fig. 4g) did not indicate critical movement of the architectures around the active site, and further docking calculations for wild type PECPS complexed with 4-hydroxyphenylpropionyl-CoA predicted that the side chain of Asn199 could be essential for ligand binding via hydrogen-bond interactions (Fig. 5 and

Supplementary Fig. 26), suggesting that the relatively reduced activity of the N199L and N199F mutants was due to the hydrophobicity or bulkiness caused by the substitutions of Asn199 to Leu and Phe. These observations suggested that PECPS utilizes only the main catalytic pocket to generate the final products.

Contrary to CUS, PECPS lacks the Ser351-Asn142-$H_2O$-Tyr207-Glu202 rearrangement, which is thought to be involved in thioester bond cleavage between the catalytic center's Cys174 and the Cys-tethered intermediate to produce a 4-coumaroyldiketide acid intermediate in CUS[57], neighboring the catalytic Cys174 at the active-site center. This phenomenon is basically caused by substitutions of Ser351, Asn142, and Tyr207 of CUS with Phe340, Ser134, and Asn199 in PECPES, respectively (Supplementary Fig. 27). This observation, especially the substitution of Ser351 of CUS with Phe340 neighboring the catalytic Cys166 in PECPS, suggests that PECPS may employ another mechanism from CUS to generate the diketide acid intermediate. Interestingly, the 4-hydroxyphenylpropionyl-$\beta$-diketide-CoA and 4-hydroxyphenylpropionyl-$\beta$-diketide acid intermediates could be captured from the reaction mixtures by LC–MS (Supplementary Fig. 28), and the diketide-CoA intermediate could be nonenzymatically hydrolyzed to yield the corresponding diketide acid in the reaction buffer (Supplementary Fig. 29). Furthermore, we found that PECPS could not produce a benzoyl-$\beta$-diketide-CoA or benzoyl-$\beta$-diketide acid intermediate from the condensation of benzoyl-CoA and one molecule of malonyl-CoA (Supplementary Fig. 30). Notably, ITC analysis revealed that the binding affinity of PECPS to 4-hydroxyphenylpropionyl-CoA ($K_D = 18.57 \pm 1.37 \mu M$) was dramatically higher than that to benzoyl-CoA ($K_D$ value is close to 1000 $\mu M$), suggesting that PECPS could use 4-hydroxyphenylpropionyl-CoA and malonyl-CoA to generate the corresponding diketide-CoA intermediate but could not form the corresponding diketide acid intermediate from thioester bond cleavage between the catalytic center's Cys166 and the Cys-tethered intermediate. From the results mentioned above, a plausible mechanism for the formation of the $C_6$–$C_5$–$C_6$ scaffold was proposed (Fig. 5).

PECPS first accepts 4-hydroxyphenylpropionyl-CoA as the starter substrate to form a 4-hydroxyphenyl-propionyl-$\beta$-diketide-CoA intermediate, from the decarboxylative condensation of 4-hydroxyphenylpropionyl-CoA and malonyl-CoA. However, due to the steric constriction of the active site architecture, PECPS terminates further polyketide chain extension reactions. Therefore, the diketide-CoA intermediate is released from the catalytic cavity, which consequently allows PECPS to tether a benzoyl unit onto the catalytic center Cys166. The subsequent tail-to-tail decarboxylative condensation of the enzyme-tethered benzoyl unit and the 4-hydroxy-phenylpropionyl-$\beta$-diketide acid, produced from the initially released diketide-CoA via non-enzymatic hydrolysis (or in vivo by an unidentified thioesterase), generates 5-(4-hydroxyphenyl)-1-phenylpentane-1,3-dione with the $C_6$–$C_5$–$C_6$ scaffold.

To confirm that a $\beta$-ketone acid could be involved in the formation of the $C_6$–$C_5$–$C_6$ scaffold, chemically synthesized 4-hydroxyphenylpropionyl-$\beta$-diketide acid (Supplementary Fig. 31) was incubated with benzoyl-CoA. The reaction expectedly generated the $C_6$–$C_5$–$C_6$ scaffold of 2 (Supplementary Fig. 32a). Additionally, PECPS could accept 3-hydroxybenzoyl-CoA or 4-hydroxybenzoyl-CoA as a starter to perform condensation with 4-hydroxyphenylpropionyl-$\beta$-diketide acid to produce the $C_6$–$C_5$–$C_6$ scaffold (Supplementary Fig. 32b, c). In contrast, 2-hydroxybenzoyl-CoA was not accepted by PECPS (Supplementary Fig. 32d). Interestingly, when PECPS was incubated with benzoyl-$\beta$-diketide acid and phenylpropionyl-CoA, the $C_6$–$C_5$–$C_6$ scaffold of 3 could also be produced (Supplementary Figs. 33–34 and Supplementary Table 1). Notably, both benzoyl-CoA and benzoyl-$\beta$-diketide acid are the key intermediates involved in the biosynthesis of salicylic acid[58–60], a signaling molecule that plays a crucial role in plant defense. Further studies to clarify the biosynthetic relationship between PECs and salicylic acid would be helpful to understand the mechanism of agarwood formation.

In summary, we have identified a diarylpentanoid-producing PKS from A. sinensis. The further successful conversion of fluorine-labeled diarylpentanoid to structurally diverse PECs demonstrated that PECs in agarwood are biosynthesized from a common $C_6$–$C_5$–$C_6$ precursor. Transient expression of PECPS in N. benthamiana and knockdown of the expression of PECPS in A. sinensis calli demonstrated that PECPS plays an important role in the biosynthesis of PECs. Crystal structure analyses suggested that PECPS utilizes unique one-pot catalytic machinery that includes the release of a diketide-CoA intermediate, which has never been reported in other known type III PKSs. The discovery of the role of PECPS in the formation of the key intermediate of the PEC biosynthetic pathway in agarwood provides insight into the overproduction of these pharmaceutically important molecules. In addition, the findings will contribute to the further exploration of the mechanism of agarwood formation and thus might be beneficial for the protection of the ecology of agarwood-producing plants, which have been seriously destroyed by overharvesting and logging.

## Methods

**General experimental procedures.** Chemicals and reagents were purchased from Sigma Aldrich (St. Louis, MO, USA) and J&K Scientific Ltd. (Beijing, China), unless noted otherwise. Restriction enzymes, DNA polymerase, and DNA ligase were purchased from Takara Biotechnology Co. Ltd. (Dalian, China). Primer synthesis and DNA sequencing were performed at Invitrogen (Shanghai, China). Malonyl-CoA was purchased from Sigma Aldrich. Chemical shifts ($\delta$) were recorded with reference to solvent signals ($^1$H NMR: $CDCl_3$ 7.26 ppm, $CD_3OD$ 3.31 ppm, acetone-$d_6$ 2.05 ppm, DMSO-$d_6$ 2.50 ppm; and $^{13}$C NMR: $CDCl_3$ 77.16 ppm, $CD_3OD$ 49.00 ppm, acetone-$d_6$ 29.92 and 206.27 ppm, DMSO-$d_6$ 40.80 ppm). HRESI-MS analyses were performed with an LCMS-IT-TOF system (Shimadzu, Kyoto, Japan) equipped with an electrospray ionization source.

4-Hydroxyphenylpropionyl-CoA, benzoyl-CoA, 2-hydroxybenzoyl-CoA, 3-hydroxybenzoyl-CoA, 4-hydroxybenzoyl-CoA, phenylpropionyl-CoA, and $p$-coumaroyl-CoA were prepared according to the reported methods[61,62] of which the details were included in Supplementary Information (Supplementary Method 1). Benzoyl-$\beta$-diketide acid and 4-hydroxyphenylpropionyl-$\beta$-diketide acid was synthesized using a similar method reported by Peuchmaur[63], and the experimental details were also included in Supplementary Information (Supplementary Method 2). Small amount of 4-hydroxyphenylpropionyl-$\beta$-diketide-CoA was enzymatically synthesized by diketide-CoA synthase (DCS) from the condensation of 4-hydroxyphenylpropionyl-CoA and malonyl-CoA[64].

**Plant tissue culture.** Fresh young leaves of A. sinensis (collected from Zhongshan, Guangdong Province, China) were cut into pieces, surface-sterilized with a sodium hypochlorite solution (2.5%) for 10 min followed by 70% ethanol for 30 s, and then washed with sterile distilled water 4 times. The treated leaf pieces were inoculated aseptically onto Murashige-Skoog (MS) medium containing 2 µg/mL naphthalene-1-acetic acid (NAA) and 1 µg/µL 6-benzylaminopurine (6-BA). After incubation at 25 °C for one month in the dark, calli were subcultured each month onto fresh MS medium containing 2 µg/mL NAA, 1 µg/mL 6-BA, 1 µg/mL dichlorophenoxyacetic acid (2,4-D) and 1 µg/mL kinetin (KT).

For the suspension cell culture, robustly growing calli on solid MS medium were transferred into 100 mL of liquid MS medium containing the same components as in the solid medium, and reciprocally shaken at 200 rpm and 25 °C in the dark. Suspension cells were subcultured every three weeks.

**Gene cloning, expression and enzyme purification.** Total RNA was isolated from 100 mg of A. sinensis calli (treated with 150 mM NaCl) using Total RNA Purification Reagent (Norgen, Canada) according to the manufacturer's instructions. The first cDNA strand was synthesized from approximately 1 µg of total RNA using oligo d(T)$_{18}$ primer and M-MLV transcriptase (Promega, Madison, WI, USA) at 42 °C for 90 min. The full-length cDNA of PECPS was obtained using 5'- and 3'-terminal PCR primers: 5'-CGC GGA TCC ATG GCA GCC AAC CCT GTG GAG TGG GTG-3' (the BamH I site is underlined), and 5'-CCC AAG CTT CTA TGC GTC TGT GAG CGT TGC AGT AG-3' (the Hind III site is underlined). The obtained PECPS cDNA contains a 1194 bp ORF encoding a polypeptide of 398 amino acids with a calculated molecular mass of 43 kDa.

The amplified DNA was digested with BamH I/Hind III and cloned into the BamH I/Hind III site of pET28a. The recombinant PECPS contains an additional hexahistidine tag at the N-terminus. After sequence confirmation, pET28a encoding wild type PECPS was introduced into the E. coli BL21(DE3) host. A single colony harboring the plasmid was inoculated into 10 mL of Luria-Bertani (LB) liquid medium containing kanamycin (50.0 µg/mL) and grown at 37 °C overnight to prepare a seed culture. Afterwards, a 2 mL aliquot of the seed culture was

transferred into 1 L of LB liquid medium and incubated at 37 °C to an OD$_{600}$ of 0.6. After the addition of 1.0 mM isopropyl-$\beta$-D-thiogalactoside to induce protein expression, the culture was further incubated at 20 °C for 20 h.

For the enzyme assay, the *E. coli* cells were harvested by centrifugation and resuspended in 40 mM potassium phosphate buffer (KPB) (pH 7.9, containing 100 mM NaCl and 5 mM imidazole). Cells were lysed by sonication, and the lysate was centrifuged at 12,000×$g$ at 4 °C for 20 min. The supernatant containing wild type PECPS was collected and passed through a Ni-NTA His-Bind™ Resin column (CWBIO, Beijing, China) containing Ni$^{2+}$ as an affinity ligand. After washing with 20 mM KPB (pH 7.9, containing 500 mM NaCl and 40 mM imidazole), the recombinant wild type PECPS was finally eluted with 15 mM KPB (pH 7.5, containing 10% glycerol and 500 mM imidazole). The purified PECPS was used in the enzyme assay after desalination and concentration with Amicon Ultra-15 centrifugal concentrators (Millipore, MI, USA). Protein concentrations were determined by the BCA method (Protein Assay, BIOMIGA) with bovine serum albumin as the standard.

For crystallization, the *E. coli* cells were harvested by centrifugation and resuspended in buffer A [20 mM Tris-HCl (pH 8.0), containing 100 mM NaCl and 10% glycerol]. The cells were disrupted by sonication and the lysate was centrifuged at 12,000 × $g$ and 4 °C for 30 min. The supernatant containing the crude fusion protein was collected and loaded onto Ni-NTA His-Bind™ Resin (CWBIO, Beijing, China) equilibrated with buffer A. After washing the resin with buffer B (buffer A containing 50 mM imidazole), the recombinant wild type PECPS was eluted with buffer C (buffer A containing 500 mM imidazole). The eluted fraction was then equilibrated to buffer C, using Macrosep® Advanced Centrifugal Device-30K units (PALL Corporation, NY, USA) by centrifugation at 8000 rpm, and applied to a Resource Q column (Cytiva, USA) equilibrated with buffer D [50 mM HEPES-NaOH (pH 7.0), containing 20 mM NaCl and 2 mM DTT]. The wild type PECPS was subsequently eluted using a linear gradient of 20–1000 mM NaCl in buffer E (buffer D containing 1.0 M NaCl). After volume reduction of the fractions containing recombinant wild type PECPS to 5 mL, the protein solution was further purified to homogeneity by gel-filtration chromatography on a Superdex 200 HiLoad 16/60 prep grade column (Cytiva, USA) equilibrated with buffer F [20 mM HEPES-NaOH (pH 7.0), containing 100 mM NaCl and 2 mM DTT]. The wild type PECPS fractions were concentrated to 15 mg/mL in buffer F using Macrosep® Advanced Centrifugal Device-30K units and used for crystallization. The protein concentration was determined by the same method as mentioned above.

**Transient expression of PECPS in *N. benthamiana*.** The open reading frame (ORF) of PECPS was amplified and inserted into the binary pCAMBIA1300-35S vector to construct the plasmid pCAMBIA1300-35S-*PECPS*. The plasmid was transformed into *Agrobacterium tumefaciens* EHA105, and the positive colony was then inoculated in LB media containing 50 ng/μL kanamycin and 30 ng/μL rifampicin and grown overnight at 28 °C. After the OD$_{600}$ value of the culture was adjusted to 0.5 with induction media (containing 10 mmol/L MES, 10 mmol/L MgCl$_2$, and 150 mmol/L acetosyringone), the culture was incubated at 28 °C for 3–4 h. After that, the culture was infiltrated into 5–8 week-old *N. benthamiana* leaves (*A. tumefaciens* containing pCAMBIA1300-35S-P19 was also infiltrated to enhance the expression of PECPS). After 2–3 days, the expression of PECPS in the *Agrobacterium*-transformed leaves was analyzed by Western blots with anti-PECPS antibody (rabbit) and anti-GAPDH antibody (mouse). For the infiltration of substrate, benzoyl-CoA dissolved in induction media (12 μmol/L) was infiltrated into *N. benthamiana* leaves (2–3 days later than the infiltration of PECPS). All *Agrobacterium*-transformed leaves were harvested after 5 days of growth in a chamber and extracted with methanol. After removal of chlorophyll by a solid phase extract cartridge, metabolites in the extract were carefully analyzed by UPLC system combined with a Thermo Fisher Scientific high resolution Q Exactive Orbitrap mass spectrometer equipped with heated electrospray ionization source. The Acquity HSS T3 Column (2.1 mm I.D. × 100 mm, 1.7 μm, Waters Corporation, Wexford, Ireland) was used. Two eluents, 0.1% formic acid (A) and acetonitrile (B) were used in a gradient program: 0–5 min, 5–35% (B); 5–20 min, 35–80% (B); 20–25 min, 80–100% (B); 25–30 min, 100% (B). The flow rate was 0.3 ml/min and the injection volume was 2 μl.

**Knockdown of *PECPS* expression in *A. sinenesis* calli.** The 200 bp fragment of *PECPS* was amplified and inserted into pBWA(V)HS-RNAi to generate intron-containing hairpin RNA constructs. The resulting recombinant vector was transformed into *A. tumefaciens* GV3101, and the positive colony harboring the recombinant plasmid was inoculated in 10 mL of LB media (containing 50 ng/μL kanamycin and 30 ng/μL rifampicin) and grown at 28 °C until the OD$_{600}$ = 0.6–0.8. After centrifugation, the *Agrobacterium* cells were resuspended in MS media for the infection of healthy *A. sinensis* calli. The infected calli were cultured on MS plates containing 300 μg/mL cefotaxime and 50 μg/mL hygromycin at 25 °C in the dark for 8 days (the expression level of *PECPS* was determined by qRT–PCR) and then incubated on new MS plates containing 150 mM NaCl, 50 μg/mL hygromycin and 300 μg/mL cefotaxime for 10 days. PECs in the treated calli were extracted and analyzed by LC–MS. The structures of PECs were tentatively assigned by their MS data recorded on Thermo Fisher Scientific high resolution Q Exactive Orbitrap mass spectrometer. The relative content of PECs were quantified using Agilent Technologies 6460 triple quadrupole mass spectrometer equipped with Acquity

HSS T3 Column (2.1 mm I.D. × 100 mm, 1.7 μm), 1,5-diphenylpentane-1,3-dione was implemented as the internal standard (**IS**). Two eluents, 0.1% formic acid (A) and acetonitrile (B) were used in a gradient program: 0–20 min, 40–90%(B). The flow rate was 0.3 ml/min and the injection volume was 5 μl.

**Isothermal titration calorimetry experiments.** ITC measurements of affinity ($K_D$), stoichiometry ($n$) and apparent enthalpy change ($\Delta H°$) were obtained using a Nano LV ITC calorimeter (TA Instruments). Samples for cell and injectant solutions were prepared in the same buffer (100 mM KPB, pH 7.5). All experiments were conducted at 37 °C with a stirring speed of 240 rpm and a sequence of 20 injections of 2.5 μL and 380 s delays between succeeding injections. The sample cell was filled with 47–85 μM protein solution (300 μl), while 1.09 mM 4-hydroxyphenylpropionyl-CoA (or 1.15 mM benzoyl-CoA) solution was placed in the syringe. The resulting data were processed with the standard NanoAnalyze software package (version 3.11.0), assuming the model of one type of binding site with an apparent stoichiometry close to 1:1. The first injection resulting in a heat change was disregarded in the later analysis. Titration experiments were performed in triplicate to show reproducibility. Control experiments confirmed that the heats of dilution caused by the reactant being titrated into buffer and the buffer into protein solution were negligible. The heats of dilution for the sample in the cell were subtracted from the final trace before integration with respect to time. The thermodynamic data obtained from the traces were averaged across three runs, and a standard deviation was calculated. Representative ITC thermograms and isotherm plots for experiments can be found in Supplementary Fig. 35.

**Site-directed mutagenesis and mutant enzymes purification.** Plasmids expressing the PECPS mutants (A210E, F340W, N199L, and N199F) were generated by the Fast Mutagenesis System (TRANSGEN BIOTECH) according to the manufacturer's instructions, using designed primers (Supplementary Table 5). The mutant enzymes were expressed, extracted, and purified by the same procedure as that for the wild type PECPS, and used in the enzyme assay and for crystallization.

**Enzyme reaction.** The standard reaction mixture contained 26 nmol of starter-CoA (4-hydroxyphenylpropionyl-CoA/benzoyl-CoA/phenylpropionyl-CoA/2-hydroxybenzoyl-CoA/3-hydroxybenzoyl-CoA/4-hydroxybenzoyl-CoA), 29 nmol of malonyl-CoA/benzoyl-$\beta$-diketide acid/4-hydroxyphenylpropionyl-$\beta$-diketide acid (note: before $\beta$-diketide acids were added to the reaction buffer, they were first dissolved in dimethyl sulfoxide to a final concentration of 2.9 mM), and 0.4 nmol of the purified recombinant enzyme, in 100 mM KPB (pH 7.5) for a final volume of 500 μL. Incubations were performed at 37 °C for 12 h and terminated by the addition of 20 μL of 20% HCl. The reaction solution was then extracted three times with 800 μL of ethyl acetate. The organic layers were combined, evaporated to dryness with nitrogen, and dissolved in 50 μL of MeOH for LCMS-IT-TOF (Shimadzu, Japan) analysis. The HPLC was equipped with an Agilent Eclipse XDB C$_{18}$ column (250 × 4.6 mm I.D., 5 μm) and eluted at a flow rate of 1.0 mL/min. For the standard assay, gradient elution was performed with H$_2$O and acetonitrile: 0–10 min, 30% acetonitrile; 10–20 min, 30–60% acetonitrile; 20–25 min, 60–70% acetonitrile; 25–30 min, 70–80% acetonitrile; 30–35 min, 80–100% acetonitrile; 35–40 min, 100% acetonitrile.

For the large scale reaction: 100–200 μmol of starter-CoA, 100–200 μmol of extenders (malonyl-CoA and another starter-CoA or $\beta$-diketide acid) and 0.4–0.6 μmol of recombinant enzyme were dissolved in 250 mL of 100 mM potassium phosphate buffer (pH 7.5) (note: before the $\beta$-diketide acids were added to the reaction buffer, they were first dissolved in dimethyl sulfoxide), and incubated at 37 °C for 12 h. The reaction mixture was stopped by the addition of 10 mL of 20% HCl and extracted with 750 mL of ethyl acetate. After solvent removal under reduced pressure, the residue was dissolved in 1 mL of MeOH and purified by semipreparative HPLC using an Agilent ZORBAX Eclipse XDB C$_{18}$ column (250 × 4.6 mm I.D., 5 μm). The products were analyzed by HRESI MS and NMR.

For the mutagenesis study, the reaction mixture contained 26 nmol of 4-hydroxyphenylpropionyl-CoA, 29 nmol of malonyl-CoA, and 0.4 nmol of the purified recombinant mutant enzyme in 100 mM KPB (pH 7.5), with a final volume of 500 μL. Incubations were performed at 37 °C for 3 h and terminated by the addition of 20 μL of 20% HCl. The reaction solution was then extracted three times with 500 μL of ethyl acetate. The combined organic layers were evaporated and dissolved in 50 μL of MeOH, for analysis with an Agilent 1260 Infinity HPLC system (Agilent Technologies, Japan). The HPLC was equipped with a Tosoh TSK-gel ODS-80Ts column (150 mm × 4.6 mm I.D., 5 μm) and eluted at a flow rate of 0.5 mL/min. For the standard assay, gradient elution was performed with H$_2$O and acetonitrile, both containing 0.1% TFA: 0–10 min, 30% acetonitrile; 10–20 min, 30–60% acetonitrile; 20–25 min, 60–70% acetonitrile; 25–30 min, 70–80% acetonitrile.

**Capture of $\beta$-diketide-CoA and $\beta$-diketide acid intermediates.** A solution of 26 nmol of 4-hydroxyphenylpropionyl-CoA, 29 nmol of malonyl-CoA, and 0.4 nmol of the purified recombinant enzyme, in 100 mM KPB (pH 6.5) with a final volume of 500 μL was incubated at 37 °C for 5 h. The reaction mixture was centrifuged at 13,000 rpm for 20 min, and a 5 μL aliquot of the supernatant was injected into an LC–MS system (ionization source: both positive and negative polarities were

applied simultaneously; column: Shim-pack XR-ODS II, 100 mm × 2.0 mm I.D.); mobile phase: 5 mM ammonium formate and acetonitrile in the following gradient program: 0–10 min, 5% acetonitrile; 10–20 min, 5–100% acetonitrile; flow rate: 0.2 mL/min).

**Chemical synthesis of fluorine-labeled substrate (F-2)**. To a solution of 4-fluorobenzaldehyde (30 mmol) in DMF (10 mL), 1-phenylbutane-1,3-dione (10 mmol), $B_2O_3$ (16 mmol), $B(OCH_3)_3$ (7 mmol), and $n$-BuNH$_2$ (8 mmol) were sequentially added. The solution was stirred at 40 °C for 12 h. Afterwards, 10 mL of 5% acetic acid was added to stop the reaction, and the precipitate was filtered and repeatedly washed with $d$H$_2$O to obtain a yellow residue (1.15 g). The residue was further separated and purified by semipreparative HPLC (column: YMC-Pack C$_{18}$, 250 mm × 10 mm I.D., 5 μm; mobile phase: 45% acetonitrile) to afford 5-(4-fluorophenyl)-1-phenylpent-4-ene-1,3-dione (810 mg). 5-(4-fluorophenyl)-1-phenylpent-4-ene-1,3-dione was dissolved in 15 mL of MeOH, and after 100 mg of 10% Pd/C was added to the solution, it was stirred at 40 °C for 4 h under flowing hydrogen gas. The reaction mixture was filtered and concentrated under reduced pressure, and the residue was separated and purified by semipreparative HPLC (column: YMC-Pack C$_{18}$ 250 mm × 10 mm; mobile phase: 49% acetonitrile) to afford F-2 (600 mg). F-2: ESI MS: $m/z$ 271 [M + H]$^+$; $^1$H NMR (400 MHz, in CDCl$_3$): δ 2.73 (2H, $t$, $J = 8.0$ Hz), 2.99 (2H, $t$, $J = 8.0$ Hz), 6.12 (1H, s), 6.98 (2H, $t$, $J = 8.0$ Hz), 7.19 (2H, $t$, $J = 8.0$ Hz), 7.42 (2H, m), 7.45 (1H, m), 7.85 (2H, $d$, $J = 8.0$ Hz). The purity of F-2 was > 98% determined by HPLC.

**Conversion of F-2 into fluorinated PECs**. Robustly growing suspension cells (200 mL × 10) were treated with NaCl (final concentration 150 mM) and immediately fed the fluorine-labeled substrate F-2 (20 mg × 10). After 10 days of culture at 200 rpm and 25 °C in the dark, the cells (wet cells, 350 g) were harvested by centrifugation and extracted with methanol (350 mL), while the liquid medium (2000 mL) was extracted with ethyl acetate (2000 mL × 3). The methanol and ethyl acetate extracts were combined, and the organic solvent was removed under reduced pressure at 42 °C to obtain a dried residue (7.08 g). Fluorinated PECs could be detected from the crude residue by LC–MS (Supplementary Fig. 8). Accordingly, the residue was separated on a silica gel column eluted with a $CH_2Cl_2$-MeOH gradient (from 20:1 to 2:1). After LC–MS analysis, the fractions that contained fluorinated PECs were further separated and purified by semipreparative HPLC (column: YMC-Pack C$_{18}$ 250 mm × 10 mm, 65% MeOH) to afford compounds C01 (3 mg), C02 (5.5 mg), C03 (3 mg), C04 (5 mg), C05 (1.5 mg), C06 (2.8 mg) and very small amounts (no more than 0.1 mg) of compounds C07–C10.

**Crystallization and structure determination**. Crystals with lengths of approximately 100 μm were obtained from a 15 mg/mL solution of the purified, recombinant wild type PECPS protein at 20 °C, in 100 mM Tris-HCl (pH 8.5) containing 0.12 M KF, 4% butanediol, and 24% PEG 8000 by the sitting-drop vapor-diffusion method. Diffraction quality crystals of the PECPS A210E mutant were obtained under the same reservoir conditions, except 15% PEG 8000 was used. Diffraction quality crystals of the PECPS F340W mutant were obtained under the same reservoir conditions, except for the use of 0.24 M KF and 24% PEG 8000. Diffraction-quality crystals of the PECPS N199L and N199F mutants were obtained under the same reservoir conditions, except for the use of 0.3 M KF and 25% PEG 8000. All crystals were transferred for 10 s into a cryoprotectant solution, consisting of each crystallization solution with 20% (v/v) glycerol and then flash-cooled at −173 °C in a nitrogen-gas stream. Diffraction data of the wild type PECPS and the PECPS A210E, F340W, and N199F mutant crystals were collected on beamline BL-1A at the Photon Factory (PF) (Tsukuba, Japan). Diffraction data of the PECPS N199L mutant crystals were collected on beamline BL-5A at the PF. These diffraction data were processed and scaled with XDS[65] and AIMLESS in the CCP4 suite[66]. The initial phases of wild type PECPS and its mutant structures were determined by molecular replacement with the crystal structures of OsPKS (PDB entry 4YJY) and wild type PECPS (PDB entry 7FFA) as search models, respectively, using phaser[67] in the CCP4 suite. The structures were modified manually with Coot[68] and refined with phenix.refine[69]. The final crystal data and intensity statistics are summarized in Supplementary Table 6. A structural similarity search was performed with the Dali program[70]. The cavity volume was calculated by CASTP (http://cast.engr.uic.edu/cast/). All crystallographic figures were prepared using PyMOL (DeLano Scientific, http://www.pymol.org). The coordinates and the structure factor amplitudes for the wild type PECPS and PECPS A210E, F340W, N199L, and N199F mutant structures have been deposited in the Protein Data Bank, with entry codes 7FFA, 7FFC, 7FFI, 7FFH, and 7FFG, respectively.

**Docking model of PECPS complexed with the intermediates and product**. AutoDock Vina (version 1.1.2)[71] was used in this project to conduct molecular docking for all calculations. The modeled protein molecules are wild type PECPS coded in the PDB ID: 7FFA. The three-dimensional structures of all ligands (malonyl-CoA, 4-hydroxyphenylpropionyl-$\beta$-diketide-CoA, phenylpropionyl-$\beta$-diketide acid, and $C_6$−$C_5$−$C_6$ scaffold product) and Cys-tethered 4-hydroxyphenylpropionyl- and benzoyl-monoketides (4-hydroxyphenylpropionyl-$L$-cysteine and benzoyl-$L$-cysteine) were generated from SMILES strings with primary optimization using the eLBOW program[72] available in PHENIX[73]. Further

optimization of the ligand was achieved through the AutoDockTools (ADT) program[74]. $S$-sulfinocysteine in the 166 residue of the wild type PECPS structure was appropriately manually converted into $L$-cysteine, 4-hydroxyphenylpropionyl-$L$-cysteine, and benzoyl-$L$-cysteine with Coot[68]. Water molecules around the cavity pocket were initially removed from the wild type PECPS structure. The protein and ligand files were converted into the PDBQT-file format by adding hydrogen atoms and partial atomic charges with the standard setting in the ADT program. For all the docking studies, Arg60, Lys64, Ser134, Asn163, Cys166 (or appropriate intermediate), Glu194, Leu198, Asn199, Thr213, Leu216, Phe217, Ile256, Leu265, Leu269, Ile273, Met308, Asn338, Phe340, and Met139 (from the paired monomer molecule) were set as the flexible side chains. A grid box size of 35 Å × 35 Å × 30 Å, covered with the flexible set residues and the cavity pocket, was used. The docking calculation was performed with the default setting, and the quality of ligand positioning in the active sites of PECPS was characterized by the free binding energy (Supplementary Table 7).

**Reporting summary**. Further information on research design is available in the Nature Research Reporting Summary linked to this article.

## Data availability

Data supporting the findings of this work are available within the paper and its Supplementary Information files. A reporting summary for this article is available as a Supplementary Information file. The structures of wild PECPS and PECPS A210E, F340W, N199L, and N199F have been deposited in the Protein Data Bank under codes 7FFA, 7FFC, 7FFI, 7FFH, and 7FFG, respectively. The GenBank accession number for the nucleotide sequence of PECPS is MH885494. RNA-Seq data that support the findings of this study have been deposited in the National Center for Biotechnology Information (NCBI) Sequence Read Archive (SRA) with accession number S3. The source data underlying Figs. 1, 3a, c, d, 4i, Supplementary Figs. 7, 20, 21, 29, and 35 are provided as a Source Data file. Source data are provided with this paper.

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

## Acknowledgements

This work was financially supported by the National Key R&D Program of China (2019YFC1711008) and the National Natural Science Foundation of China (81773842) and partially supported by Grants-in-Aid for Scientific Research from the Ministry of Education, Culture, Sports, Science and Technology, Japan (JP19H04649 and 20K16025).

## Author contributions

H.M., P.F.T., I.A. and S.P.S. designed the experiments. X.H.W. and B.W.G. performed the tissue culture, gene cloning, protein purification, enzyme reactions, and biotransformation. C.P.W. and Q.Q.L. prepared the protein crystals and performed crystal data collection. T.K., T.M. and Y.E.L. performed protein structure refinements and analysis. Y.N. performed docking simulation. Z.X.Z. performed the qRT-PCR, western blot analysis, and bio-transformation experiments. Z.K.Z. and B.W.Q. performed the LC–MS analysis. J.W. synthesized the substrates used in this study. J.L., X.L. and Y.N. were involved in the data analysis and discussion of the results. S.P.S. and H.M. wrote the manuscript.

## Competing interests

The authors declare no competing interests.
