## [Peer Review File · Nature Communications]

Title: Identification of a Diarylpentanoid-Producing Polyketide Synthase Revealing an Unusual Biosynthetic Pathway of 2-(2-Phenylethyl)chromones in AgarwoodREVIEWER COMMENTS

Reviewer #1 (Remarks to the Author):

In the manuscript "Identification of a Diarylpentanoid-Producing Plant Polyketide Synthase Revealing an Unusual Biosynthetic Pathway of 2-(2-Phenylethyl)chromones in Agarwood," Wang et al. investigated the chemical and molecular mechanism of the 2-(2-Phenylethyl)chromones (PECs) biosynthesis pathway in Agarwood. Since PECs have potential value for various pharmacological applications, understanding the biosynthesis and catalytic mechanisms of a core enzyme diarylpentanoid-producing polyketide synthase (PECPS) from *Aquilaria sinensis* is able to contribute to the research field of biotechnology significantly.

To understand the catalytic mechanism of PECPS, the authors provided the crystal structures of WT PECP (in an apo form) and 4 point-mutants, which are A210E, F340W, N199L, & N199F. Using a combination of analytical and structural analysis approaches, the authors proposed a mechanism for the formation of the C6-C5-C6 scaffold. Here are some comments to improve this manuscript.

Page 6. & Fig. 1E

- Please provide the number of samples. Can you provide the PECPS protein expression levels of unstressed *A. sinensis* calli at the same time points (as a control)? Can you quantify the PECPS protein expression levels?

Page 7. & Fig. 3 (Size of the ligand-binding cavity)

- The authors calculated the sizes of the ligand-binding cavities using CASTP (<http://cast.engr.uic.edu/cast/>) and rationalized their hypothesis using differences in cavity volume. The result can be valid and can be used to explain the different activities. However, the authors should be more careful to compare the volume of each ligand-binding cavity directly. Because the crystal structure is a snapshot of a dynamic protein, the volume of a cavity can vary (even in the same protein) depending on the state of the protein (e.g., apo form vs complexed forms; or open-form vs closed-form). For example, the authors used an apo form of PECPS (Fig. 3a) to calculate the volume. However, they did not mention the states of CUS and MsCHS. Please provide more information on the structure models of CUS and MsCHS (e.g., PDB codes). Besides, although the CASTP server calculated the sizes (e.g., 247 Å³ and 754 Å³) of the cavities, the difference does not appear effectively in the current view. Please find a better angle and provide a supplemental figure to show the different volumes of cavities.

Fig. 3, Fig. 4, & Pages 17-18 (Docking experiment)

- Since most of the PECPS crystal structures are the apo form (Supplementary Table 5). The authors conducted docking experiments to find the key amino acid residues in the active site. However, as described in Methods, the authors manually docked the ligands into the active site using Coot, a structure building program rather than a computational docking program. As mentioned above, the crystal structure is a snapshot of a dynamic protein. The side chains in the active site can be flexible for

binding. For example, if Asn199 of PECPS participates in the ligand binding, then the side chain of Asn199 can be shifted toward the ligand, the "Pocket C" cavity can be generated. Also, manual docking cannot predict all possible conformers. Please use computational docking software/server and provide predicted docking results, such as predicted structure and binding affinity.

Fig. 3i (Mutants & Enzyme assays)

- The authors generated 4 point-mutants and analyzed the enzymatic activity using the HPLC chromatograms. The HPLC-based enzyme assays could be time-consuming and tedious. However, to understand the enzyme mechanism, a basic steady-state kinetic analysis is required. If the authors want to compare the relative activity (%) of each mutant, at least, values of the catalytic efficiency (k_{cat}/K_m) or maximal velocity (V_{max}) of each enzyme should be provided. Otherwise, to confirm the contribution of ligand-binding, the authors can provide experimental protein-ligand binding results using Isothermal titration calorimetry (ITC) or Surface Plasmon Resonance (SPR).

Supplementary Table 5. & Fig. 3d (Structure of PECPS A210E)

- The I/σ value of PECPS A210E is a bit low. To ensure data quality, please provide the $CC_{1/2}$ or trim the data set. Based on Supplementary Table 5. and the X-ray structure validation report, only PECPS A210E contains some ligands. Do the ligands conformationally alter the structure?

Fig. 4 (Proposed mechanism)

- As mentioned above, the structure of this ligand complex (Figs 4a-d) is a manually predicted docking model. It would be better to present a mechanism by using a more reliable model.

Overall, the topic of the study is interesting, and the quality of the experimental data is generally good. However, the structural analysis should be more reliable to support the hypothesis and additional functional studies are needed to publish it in Nature Communications.

Reviewer #2 (Remarks to the Author):

Wang et al. reported the discovery of a type III polyketide synthase (PKS)-encoding gene from Agarwood. Through in vitro enzyme assays, the authors showed that this type III PKS harbors novel diarylpentanoid-producing polyketide synthase (PECPS) activities, and is likely involved in the production of 2-(2-Phenylethyl)chromones (PECs) in Agarwood. The authors then carried out structural analysis of PECPS, and through a series of site-directed mutagenesis experiments as well as additional structures of mutant enzymes, the authors assessed the roles of a number of active-site-lining residues in catalysis. A catalytic mechanism of PECPS is proposed. The PECPS activity is novel. However, its in vivo biochemical function and its role in PEC biosynthesis in Agarwood remain undemonstrated, which dampens the overall significance of this study. My detailed comments are listed below.

Major comments:

1. Line 97, the authors referred to a previous failed attempt to characterize Aschs1, a type III PKS from Agarwood. The statement “However, further determination of the function of Aschs1 failed because the gene could not be heterologously expressed” is ambiguous. The authors should describe how and why this previous attempt failed. Line 101, there is no mention how Aspks1-2 was identified. It is unclear why PECPS was then selected over other Agarwood PKSs to be studied in this paper to be a candidate enzyme for PEC biosynthesis. How many type III PKS genes did the authors uncover from their own transcriptome analysis? These genes should all be included in the phylogenetic analyses, which should also include numerous previously described PKSs from other species. The statement “Phylogenetic analysis revealed that PECPS was grouped into the class of plant non-chalcone synthases” is confusing, as there are CHS enzymes in the so called “non-chalcone synthases” group as shown in Fig. 1c. It is critical to show the activities of some of the other highly expressed PKSs, elicited PKSs, or closely related PKSs in Agarwood, e.g. Aschs1 and Aspks1-2. This will help to clarify the role of PECPS in PEC biosynthesis in Agarwood. The phylogenetic tree is better to be displayed in rectangle format for the readers to see the relative branch length.

2. A major weakness of the paper is that the *in vivo* biochemical function of PECPS is not demonstrated. It is known that *in vitro* activities of plant type III PKSs sometimes differ from their major activities *in vivo*. To support the main claim “PECs in agarwood are biogenetically synthesized from a common C6-C5-C6 precursor produced by PECPS, confirmed that PECPS plays important role in the biosynthesis of PECs (Line 221)”, the author need to demonstrate PECPS indeed harbors the claimed activities *in vivo*. For example, this can be demonstrated by transient expression of PECPS in *Nicotiana*, and/or VIGS experiment in Agarwood callus to knockdown expression of PECPS.

3. As type III PKSs are known to harbor alternative catalytic outcomes, it is important to show the full spectrum of the products produced from various PECPS enzyme assays in Fig. 2d. The nature of the side products should also be shown in the main figure.

4. On a related note, Line 183, when certain mutants show compromised overall activity, how about the impact on their side product profiles?

5. Line 191, “Contrary to CUS, a nucleophilic water molecule activated by a hydrogen bond network was not detected in PECPS.” I believe the authors are referring to the observation of water molecules that are H-bonding with the catalytic residues seen in the CUS crystal structure. The absence of these in the PECPS structure doesn’t disapprove that nucleophilic water is not part of the PECPS catalytic mechanism, it may not be captured by the current crystal structure.

6. Line 214, “Interestingly, when PECPS was incubated with a benzoyl- β -diketide acid and phenylpropionyl-CoA, the C6-C5-C6 scaffold of 3 could also be produced (Supplementary Fig. 26 and Supplementary Table 1)”. The Supplementary Fig. 26 and Supplementary Table 1 are structural elucidation data for compound 3. Please show the actual enzyme assay data.

7. Line 220, “The further successful conversion of fluorine-labeled diarylpentanoid into structurally

diverse PECs demonstrated that PECs in agarwood are biogenetically synthesized from a common C6-C5-C6 precursor produced by PECPS, confirmed that PECPS plays important role in the biosynthesis of PECs.” This conclusion is false. The tracing experiment does not support the involvement of PECPS in PEC biosynthesis in Agarwood. Again, this speaks to the importance of demonstrating the in vivo function of PECPS as mentioned earlier.

Minor comments

1. The term “biogenetically synthesized” is used at least three times. This is a problematic term as genetic approach is a biological approach. I suggest change it to “biosynthetically” or “biosynthesized” depending on the context.
2. Line 101, when referring to a gene “e.g. (*pecps*)”, please use italic capitalized letters for the gene name.
3. Line 106, “Fortunately, the full-length cDNA of *pecps* could be easily expressed in *E. coli* as a fusion protein with a hexahistidine tag”. Please change “full-length cDNA of *pecps*” to “full-length PECPS protein”, so that the concepts of DNA and protein are not mixed.
4. When various key activity-dictating residues are discussed, it would be helpful to show them on Supplementary Fig. 2, and also summarize their roles with different colors and also in the figure legend.

Response to referees

The authors thank the editor and reviewers for their constructive comments, which helped improve the quality of this manuscript.

Reviewer #1 (Remarks to the Author):

In the manuscript "Identification of a Diarylpentanoid-Producing Plant Polyketide Synthase Revealing an Unusual Biosynthetic Pathway of 2-(2-Phenylethyl)chromones in Agarwood," Wang et al. investigated the chemical and molecular mechanism of the 2-(2-Phenylethyl)chromones (PECs) biosynthesis pathway in Agarwood. Since PECs have potential value for various pharmacological applications, understanding the biosynthesis and catalytic mechanisms of a core enzyme diarylpentanoid-producing polyketide synthase (PECPS) from *Aquilaria sinensis* is able to contribute to the research field of biotechnology significantly.

To understand the catalytic mechanism of PECPS, the authors provided the crystal structures of WT PECPS (in an apo form) and 4 point-mutants, which are A210E, F340W, N199L, & N199F. Using a combination of analytical and structural analysis approaches, the authors proposed a mechanism for the formation of the C6-C5-C6 scaffold. Here are some comments to improve this manuscript.

Page 6. & Fig. 1E

- Please provide the number of samples. Can you provide the PECPS protein expression levels of unstressed *A. sinensis* calli at the same time points (as a control)? Can you quantify the PECPS protein expression levels?

Reply: We carefully quantified the expression levels of PECPS protein in both unstressed and NaCl-treated *A. sinensis* calli (each experiment was repeated three times). We prepared a new Fig. 1 in the revised manuscript, Figures of phylogenetic tree and the relative expression level of *PECPS* in the original Fig. 1 were moved to the revised Supplementary Information and numbered as Figure 3 and Figure 7, respectively. Please see details in the revised Fig. 1c, 1d in the main text and Figure 7 in the Supplementary Information.

Page 8. & Fig. 3 (Size of the ligand-binding cavity)

- The authors calculated the sizes of the ligand-binding cavities using CASTP (<http://cast.engr.uic.edu/cast/>) and rationalized their hypothesis using differences in cavity volume. The result can be valid and can be used to explain the different activities. However, the authors should be more careful to compare the volume of each ligand-binding cavity directly. Because the crystal structure is a snapshot of a dynamic protein, the volume of a cavity can vary (even in the same protein) depending on the state of the protein (e.g., apo form vs complexed forms; or open-form vs closed-form). For example, the authors used an apo form of PECPS (Fig. 3a) to calculate the volume. However, they did not mention the states of CUS and MsCHS. Please provide more

information on the structure models of CUS and MsCHS (e.g., PDB codes). Besides, although the CASTP server calculated the sizes (e.g., 247 Å³ and 754 Å³) of the cavities, the difference does not appear effectively in the current view.

Please find a better angle and provide a supplemental figure to show the different volumes of cavities.

Reply: We agree your suggestion. The differences of the cavity size in the crystal structure are not reasons to discuss the differences between PECPS and the other type III PKSs in this study since we have obtained only its apo-structure. According to your comments, we have revised the relevant descriptions in the revised manuscript as one possibility, by providing the states of MsCHS and CUS and their PDB codes. In these revisions, we also added the cavity size of the apo structure of MsCHS (742 Å³, PDB ID: 1BI5), which showed almost the same cavity volume as that of the naringenin-complex form (754 Å³, PDB ID: 1CGK) in the CASTP server calculation. We also provided a new figure to effectively show the different volumes of the PECPS, MsCHS, and CUS cavities (Supplementary Fig. 25).

Fig. 3, Fig. 4, & Pages 21-22 (Docking experiment)

- Since most of the PECPS crystal structures are the apo form (Supplementary Table 5). The authors conducted docking experiments to find the key amino acid residues in the active site. However, as described in Methods, the authors manually docked the ligands into the active site using Coot, a structure building program rather than a computational docking program. As mentioned above, the crystal structure is a snapshot of a dynamic protein. The side chains in the active site can be flexible for binding. For example, if Asn199 of PECPS participates in the ligand binding, then the side chain of Asn199 can be shifted toward the ligand, the "Pocket C" cavity can be generated. Also, manual docking cannot predict all possible conformers. Please use computational docking software/server and provide predicted docking results, such as predicted structure and binding affinity.

Reply: We performed docking simulations using the AutoDock Vina program, where 19 side chains of the residues, including that of Asn199 around the active site, were set as flexible residues. The docking simulation predicted no critical movement of the side chains, including the apparent rotation of Asn199, even though the substrate and intermediate were located in the active site cavity. These results and procedures are provided in the "Structural basis for the PECPS catalytic mechanism" section and "Method" section as well as Supplementary Figure 27 in the revised manuscript. The free binding energies are provided as "Supplementary Table 7. In accordance with the new docking simulations, we also updated Figure 4 in the main text.

Fig. 3i (Mutants & Enzyme assays)

- The authors generated 4 point-mutants and analyzed the enzymatic activity using the HPLC chromatograms. The HPLC-based enzyme assays could be time-consuming and tedious. However, to understand the enzyme mechanism, a basic steady-state kinetic analysis is required. If the authors want to compare the relative activity (%) of each mutant, at least, values of the catalytic efficiency (k_{cat}/K_m) or maximal velocity (V_{max}) of each enzyme should be provided. Otherwise, to confirm the contribution of ligand-binding, the

authors can provide experimental protein-ligand binding results using Isothermal titration calorimetry (ITC) or Surface Plasmon Resonance (SPR).

Reply: We newly performed ITC analysis with respect to the binding ability of PECPS and its four mutant enzymes against 4-hydroxyphenylpropionyl-CoA. The results of the analysis followed the previously described results of the relative activity of the mutants against that of the wild type. We provided the ITC results in the “Structural basis for the PECPS catalytic mechanism” section and “Supplementary Table 4”. In the revised manuscript, the results and discussion for the F340W mutant, which has been constructed to discuss the role of the region corresponding to pocket A in MsCHS, were combined with relevant descriptions. We also provided the binding ability of wild type PECPS against benzoyl-CoA in the ITC experiment in the same section to support the substrate binding order of PECPS.

Supplementary Table 6. & Fig. 3d (Structure of PECPS A210E)

- The I/σ value of PECPS A210E is a bit low. To ensure data quality, please provide the CC1/2 or trim the data set. Based on Supplementary Table 5. and the X-ray structure validation report, only PECPS A210E contains some ligands. Do the ligands conformationally alter the structure?

Reply: We determined that the quality of the data is acceptable, since the CC1/2 value of the PECPS A210E structure was 0.759 in the outer shell (2.65 Å-2.61 Å) and 0.991 overall (49.13 Å-2.61 Å). We provided CC 1/2 data for all structures in Supplementary Table 6. In this structure, glycerol molecules have been attached on the protein surfaces. However, significant structural changes were not observed.

Fig. 4 (Proposed mechanism)

- As mentioned above, the structure of this ligand complex (Figs 4a-d) is a manually predicted docking model. It would be better to present a mechanism by using a more reliable model.

Reply: We updated the Figure 4 using the new simulation results.

Overall, the topic of the study is interesting, and the quality of the experimental data is generally good. However, the structural analysis should be more reliable to support the hypothesis and additional functional studies are needed to publish it in Nature Communications.

Reviewer #2 (Remarks to the Author):

Wang et al. reported the discovery of a type III polyketide synthase (PKS)-encoding gene from Agarwood. Through in vitro enzyme assays, the authors showed that this type III PKS harbors novel diarylpentanoid-producing polyketide synthase (PECPS) activities, and is likely involved in the production of 2-(2-Phenylethyl)chromones (PECs) in Agarwood. The authors then carried out structural analysis of PECPS, and through a series of site-directed mutagenesis experiments as well as additional structures of mutant enzymes, the authors assessed the roles of a number of active-site-lining residues in

catalysis. A catalytic mechanism of PECPS is proposed. The PECPS activity is novel. However, its *in vivo* biochemical function and its role in PEC biosynthesis in Agarwood remain undemonstrated, which dampens the overall significance of this study. My detailed comments are listed below.

Major comments:

1. Line 97, the authors referred to a previous failed attempt to characterize Aschs1, a type III PKS from Agarwood. The statement “However, further determination of the function of Aschs1 failed because the gene could not be heterologously expressed” is ambiguous. The authors should describe how and why this previous attempt failed. Line 101, there is no mention how Aspks1-2 was identified. It is unclear why PECPS was then selected over other Agarwood PKSs to be studied in this paper to be a candidate enzyme for PEC biosynthesis. How many type III PKS genes did the authors uncover from their own transcriptome analysis? These genes should all be included in the phylogenetic analyses, which should also include numerous previously described PKSs from other species. The statement “Phylogenetic analysis revealed that PECPS was grouped into the class of plant non-chalcone synthases” is confusing, as there are CHS enzymes in the so called “non-chalcone synthases” group as shown in Fig. 1c. It is critical to show the activities of some of the other highly expressed PKSs, elicited PKSs, or closely related PKSs in Agarwood, e.g. Aschs1 and Aspks1-2. This will help to clarify the role of PECPS in PEC biosynthesis in Agarwood. The phylogenetic tree is better to be displayed in rectangle format for the readers to see the relative branch length.

Reply: Thank you for your comments. We reorganized the related text to avoid confusion. A total of five *PKS* genes with upregulated expression were found from our transcriptomic dataset. Initially, we focused on the cloning and functional identification of *AsCHS1*, which is the gene with the most significantly upregulated expression among the five candidates. However, *AsCHS1* cloned from *A. sinensis* always contains an intron and cannot be heterologously expressed in *E. coli* even after removal of the intron. Therefore, we switched to identify the catalytic functions of the other four candidate genes. After successful expression of all four candidates in *E. coli*, the catalytic function of three candidates was identified as normal PKSs by *in vitro* enzymatic reaction; they were one chalcone-producing PKS (*AsCHS*) and two pyrone-producing PKSs (*AsPKS1* and *AsPKS2*). Please also see details in the published paper (*Biochem. Bioph. Res. Comm.* 2017, 486, 1040–1047), where *AsCHS* was characterized as chalcone synthase with the same name of *AsCHS1* (we apologize for the confusion caused by our negligence). According to the reviewer’s comments, all five candidate genes were included in amino acid sequence alignment (Supplementary Figure 2) and phylogenetic analyses (Supplementary Figure 3) with other previously described plant PKSs. As for the comment that there are CHS enzymes in the so-called “non-chalcone synthases” group, we found that a PKS named *AsCHS1* (EF103196.1) (this protein is identical to the unsuccessfully expressed *AsCHS1* mentioned in this work) was grouped in “non-chalcone synthases”; however, its catalytic function has never been identified, and “*AsCHS1*” was named by the submitter from another group when they submitted this gene to GeneBank (accession

number: EF103196.1). Related interpretations were also included in the revised Figure legend. In the revised manuscript, we also moved the phylogenetic tree from the main text to the Supplementary Information (Supplementary Figure 3).

2. A major weakness of the paper is that the *in vivo* biochemical function of PECPS is not demonstrated. It is known that *in vitro* activities of plant type III PKSs sometimes differ from their major activities *in vivo*. To support the main claim “PECs in agarwood are biogenetically synthesized from a common C₆-C₅-C₆ precursor produced by PECPS, confirmed that PECPS plays important role in the biosynthesis of PECs (Line 221)”, the author need to demonstrate PECPS indeed harbors the claimed activities *in vivo*. For example, this can be demonstrated by transient expression of PECPS in *Nicotiana*, and/or VIGS experiment in Agarwood callus to knockdown expression of PECPS.

Reply: Thank you for the constructive comments. According to the reviewer’s suggestions, we further investigated the *in vivo* biochemical function of PECPS using two methods to express PECPS in *Nicotiana benthamiana* and knockdown the expression of *PECPS* in *A. sinensis calli*. As a result, transient expression of PECPS in the *N. benthamiana* leaves resulted in the accumulation of a C₆-C₇-C₆ scaffold tentatively assigned as 5-hydroxy-1,7-bis(4-hydroxyphenyl)heptan-3-one, which was not observed in the wild type *N. benthamiana* leaves. When *Agrobacterium* harboring the *PECPS* gene and benzoyl-CoA (3 days later than the infiltration of *Agrobacterium*) infiltrated the *N. benthamiana* leaves, the expected C₆-C₅-C₆ scaffold could not be found, while it led to the accumulation of a PEC scaffold of 2-(4-hydroxyphenethyl)-4*H*-chromen-4-one, suggesting that oxidase(s) in wild type *N. benthamiana* leaves might contribute to the oxidative cyclization of the C₆-C₅-C₆ scaffold produced by PECPS to form the PEC scaffold. Further infiltration of the C₆-C₅-C₆ scaffolds, 5-(4-hydroxyphenyl)-1-phenylpentane-1,3-dione and 1,5-diphenylpentane-1,3-dione, into the wild type *N. benthamiana* leaves expectedly led to the accumulation of two PECs, which confirmed the above deduction. In contrast, the related C₆-C₇-C₆ and PEC scaffolds were not detected in the *N. benthamiana* leaves infiltrated with AsCHS, AsCHS1, AsPKS1 and AsPKS2 (data not shown). In addition, we found that the contents of PECs in the PECPS knockdown calli (treated with 150 mM NaCl) were dramatically decreased compared with those in the calli treated with 150 mM NaCl alone. The results demonstrated that PECPS plays an important role in the biosynthesis of PECs in agarwood.

The related discussions and experimental methods were included in the revised manuscript (highlighted text on pages p7, p8, p16, and p17, Figures 20–23 in the Supplementary Information).

3. As type III PKSs are known to harbor alternative catalytic outcomes, it is important to show the full spectrum of the products produced from various PECPS enzyme assays in Fig. 2d. The nature of the side products should also be shown in the main figure.

Reply: We have included the full HPLC chromatogram in the revised manuscript in Fig. 1b. However, after careful analysis of the LC–MS data, side products were not observed under the mentioned reaction conditions.

4. On a related note, Line 183, when certain mutants show compromised overall activity,

how about the impact on their side product profiles?

Reply: We did not find side products under our assay conditions. The docking simulation supported the effect of the activities in the case of Asn199 mutants. The suggestions were added to “Structural basis for the PECPS catalytic mechanism” of the main text.

5. Line 191, “Contrary to CUS, a nucleophilic water molecule activated by a hydrogen bond network was not detected in PECPS.” I believe the authors are referring to the observation of water molecules that are H-bonding with the catalytic residues seen in the CUS crystal structure. The absence of these in the PECPS structure doesn’t disprove that nucleophilic water is not part of the PECPS catalytic mechanism, it may not be captured by the current crystal structure.

Reply: Thank you very much for your suggestion. We deleted this description and mentioned the structural difference near the catalytic center between PECPS and CUS by comparing residues in this region to determine possible differences in the catalytic mechanism between PECPS and CUS. We also provided Supplementary Figure 28 comparing these structures of PECPS and CUS.

6. Line 214, “Interestingly, when PECPS was incubated with a benzoyl- β -diketide acid and phenylpropionyl-CoA, the C6-C5-C6 scaffold of 3 could also be produced (Supplementary Fig. 26 and Supplementary Table 1)”. The Supplementary Fig. 26 and Supplementary Table 1 are structural elucidation data for compound 3. Please show the actual enzyme assay data.

Reply: The HPLC chromatogram for the mention enzymatic reaction was included in the revised Supplementary Information as Figure 34.

7. Line 220, “The further successful conversion of fluorine-labeled diarylpentanoid into structurally diverse PECs demonstrated that PECs in agarwood are biogenetically synthesized from a common C6-C5-C6 precursor produced by PECPS, confirmed that PECPS plays important role in the biosynthesis of PECs.” This conclusion is false. The tracing experiment does not support the involvement of PECPS in PEC biosynthesis in Agarwood. Again, this speaks to the importance of demonstrating the *in vivo* function of PECPS as mentioned earlier.

Reply: Thank you for the constructive comments. We performed experiments to express PECPS in *Nicotiana benthamiana* and knockdown the expression of *PECPS* in *A. sinensis* calli to investigate the *in vivo* function of PECPS. The results suggested that PECPS plays an important role in the biosynthesis of PECs in agarwood. In addition, we also changed the sentence as “The further successful conversion of fluorine-labeled diarylpentanoid to structurally diverse PECs demonstrated that PECs in agarwood are biosynthesized from a common C₆-C₅-C₆ precursor. Transient expression of PECPS in *N. benthamiana* and knockdown of the expression of *PECPS* in *A. sinensis* calli demonstrated that PECPS plays an important role in the biosynthesis of PECs.”

Minor comments

1. The term “biogenetically synthesized” is used at least three times. This is a problematic

term as genetic approach is a biological approach. I suggest change it to “biosynthetically” or “biosynthesized” depending on the context.

Reply: Revised accordingly.

2. Line 101, when referring to a gene “e.g. (*pecps*)”, please use italic capitalized letters for the gene name.

Reply: Revised accordingly.

3. Line 106, “Fortunately, the full-length cDNA of *pecps* could be easily expressed in *E. coli* as a fusion protein with a hexahistidine tag”. Please change “full-length cDNA of *pecps*” to “full-length PECPS protein”, so that the concepts of DNA and protein are not mixed.

Reply: Revised accordingly.

4. When various key activity-dictating residues are discussed, it would be helpful to show them on Supplementary Fig. 2, and also summarize their roles with different colors and also in the figure legend.

Reply: We revised the Supplementary Fig.2, in which the key residues were highlighted by different color.

REVIEWER COMMENTS

Reviewer #1 (Remarks to the Author):

In the revised manuscript "Identification of a Diarylpentanoid-Producing Plant Polyketide Synthase Revealing an Unusual Biosynthetic Pathway of 2-(2-Phenylethyl)chromones in Agarwood," Wang et al. investigated the chemical and molecular mechanism of the 2-(2-Phenylethyl)chromones (PECs) biosynthesis pathway in Agarwood. The authors answered the reviewer's questions well and greatly improved the quality of the manuscript. In particular, the results of the newly added experiments (e.g., simulation and ITC) answered most of my questions. However, I have a few more questions about the new ITC results.

In general, one of the most important data in ITC is the n value (=reaction stoichiometry). In Method, the authors noted that a single-site binding model was used for processing. However, in Supplementary Table 4, the n values are inconsistent with their original fitting model. In many cases, the n values tend to vary if the concentrations (of proteins or ligands) are not accurate or the data quality is insufficient. Therefore, the authors should account for the difference in n values. In addition, based on the raw data in Supplementary Figure 36, some of the data, especially panel (b) PECPS A210E & panel (e) PECPS N199F, seem challenging to process and are likely to be inaccurate. For reference, can you provide raw data of another duplicate?

Response to referees

The authors are deeply thankful to the reviewers who have spent much of their precious time on reviewing this manuscript, providing us very thoughtful and instructive comments and suggestions which are really helpful for improving the quality of our manuscript.

Reviewer #1 (Remarks to the Author):

In the revised manuscript "Identification of a Diarylpentanoid-Producing Plant Polyketide Synthase Revealing an Unusual Biosynthetic Pathway of 2-(2-Phenylethyl)chromones in Agarwood," Wang et al. investigated the chemical and molecular mechanism of the 2-(2-Phenylethyl)chromones (PECs) biosynthesis pathway in Agarwood. The authors answered the reviewer's questions well and greatly improved the quality of the manuscript. In particular, the results of the newly added experiments (e.g., simulation and ITC) answered most of my questions. However, I have a few more questions about the new ITC results. In general, one of the most important data in ITC is the n value (=reaction stoichiometry). In Method, the authors noted that a single-site binding model was used for processing. However, in Supplementary Table 4, the n values are inconsistent with their original fitting model. In many cases, the n values tend to vary if the concentrations (of proteins or ligands) are not accurate or the data quality is insufficient. Therefore, the authors should account for the difference in n values. In addition, based on the raw data in Supplementary Figure 36, some of the data, especially panel (b) PECPS A210E & panel (e) PECPS N199F, seem challenging to process and are likely to be inaccurate. For reference, can you provide raw data of another duplicate?

Reply: We are very thankful for the reviewer's instructive comments. To improve the quality of the ITC data, we re-performed the experiments and collected the data under further optimized condition. Please see the representative ITC thermograms and isotherm plots on the next page (Fig. 1). In the Figure, the data of the N199F mutant was represented by only a scatter diagram rather than a fitting curve (Fig. 1e), since obtaining the accurate K_D value by fitting a reasonable curve was impossible, due to very weak binding affinity of this enzyme to 4-hydroxyphenylpropionyl-CoA (K_D value is close to 1000 μ M). The binding affinity values of the F340, N199L, and N199F mutants obtained by the re-performed ITC analysis were in good agreement with the previously obtained relative activities of these mutant enzymes. For the A210E mutant (Fig. 1b), although we have spent much time on the evaluation of the binding affinity of this protein, tailed peaks are always observed on its ITC thermograms. Thus, the binding affinity value of the A210E mutant was estimated to be 2.1 times lower than that of the wild-type. However, we believe that the data of the A210E mutant could support our proposed mechanism of PECPS, because if the pocket B is employed in the PECPS reaction, the K_D value of the A210E mutant to 4-hydroxyphenylpropionyl-CoA will be dramatically decreased, as observed in the N199F mutant. In accordance with these data, we revised related sentences in the revised manuscript. For reference, we also submitted the new ITC data of three duplications along

with the old one (previous submitted ITC data), please kindly see details in the submitted “ITC data—Files for Review Only”.

Fig. 1 Representative ITC thermograms and isotherm plots for the interaction of 4-hydroxyphenylpropionyl-CoA and different proteins. **a** wild PECPS; **b** PECPS A210E; **c** PECPS F340W; **d** PECPS N199L; **e** PECPS N199F. Each experiment was independently repeated three times. Fitting curve for PECPS N199F is not presented due to its very weak binding ability with 4-hydroxyphenylpropionyl-CoA (K_D value is close to 1000 μM). Data represent the mean ± SD (n = 3).

The new ITC data mentioned above were included in the revised main text and supplementary information (Supplementary Fig. 35, and Supplementary Table 4).

Reviewer #2

The manuscript was suitable for publication. Reviewer #2 suggested including representative *N. benthamiana* transient expression and *A. sinensis calli* RNAi results as

an additional main figure.

Reply: We appreciate the reviewer's comments. We included the representative N. benthamiana transient expression and A. sinensis calli RNAi results as an additional main figure (Fig. 3) in the revised manuscript.

Additional revisions by authors.

In accordance with the additional contribution for this study, the order of the authors on the manuscript has been changed. The "Author contribution" section has been also revised.

REVIEWERS' COMMENTS

Reviewer #1 (Remarks to the Author):

The authors addressed the issue (e.g., ITC data) well and improved the quality of the manuscript. This manuscript provides the reader with detailed information to understand the data.